



# Asymmetric Responses of Primary Productivity to Altered Precipitation Simulated by Ecosystem Models across Three Long-term Grassland Sites

Donghai Wu[1], Philippe Ciais[2], Nicolas Viovy[2], Alan K. Knapp[3], Kevin Wilcox[4], Michael Bahn[5], Melinda
D. Smith[3], Sara Vicca[6], Simone Fatichi[7], Jakob Zscheischler[8], Yue He[1], Xiangyi Li[1], Akihiko Ito[9], Almut
Arneth[10], Anna Harper[11], Anna Ukkola[12], Athanasios Paschalis[13], Benjamin Poulter[14], Changhui Peng[15,16],
Daniel Ricciuto[17], David Reinthaler[5], Guangsheng Chen[18], Hanqin Tian[18], Hélène Genet[19], Jiafu Mao[17],
Johannes Ingrisch[5], Julia E.S.M. Nabel[20], Julia Pongratz[20], Lena R. Boysen[20], Markus Kautz[10], Michael
Schmitt[5], Patrick Meir[21,22], Qiuan Zhu[16], Roland Hasibeder[5], Sebastian Sippel[23], Shree R.S. Dangal[18,24],
Stephen Sitch[25], Xiaoying Shi[17], Yingping Wang[26], Yiqi Luo[4,27], Yongwen Liu[1], Shilong Piao[1]

[1] Sino-French Institute for Earth System Science, College of Urban and Environmental Sciences, Peking University, Beijing, 100871, China.

[2] Laboratoire des Sciences du Climat et de l'Environnement, CEA-CNRS-UVSQ, Gif-Sur-Yvette 91191, France.

[3] Department of Biology and Graduate Degree Program in Ecology, Colorado State University, Fort Collins, CO 80523, USA.

[4] Department of Microbiology and Plant Biology, University of Oklahoma, Norman, OK 73019, USA.

[5] Institute of Ecology, University of Innsbruck, 6020 Innsbruck, Austria.

[6] Department of Biology, University of Antwerp, Universiteitsplein 1, 2610 Wilrijk, Belgium.

[7] Institute of Environmental Engineering, ETH Zurich, 8093 Zurich, Switzerland.

[8] Institute for Atmospheric and Climate Science, ETH Zurich, 8092 Zurich, Switzerland.

[9] National Institute for Environmental Studies, Tsukuba, Ibaraki 305-8506, Japan.

[10] Karlsruhe Institute of Technology, 82467 Garmisch-Partenkirchen, Germany.

[11] College of Engineering, Mathematics and Physical Sciences, University of Exeter, Exeter, EX4 4QF, UK.

[12] ARC Centre of Excellence for Climate System Science, University of New South Wales, Kensington, NSW 2052, Australia.

[13] Department of Civil and Environmental Engineering, Imperial College London, London, SW7 2AZ, UK.

[14] NASA Goddard Space Flight Center, Biospheric Sciences Laboratory, Greenbelt, MD 20771, USA.

[15] Institute of Environment Sciences, Biology Science Department, University of Quebec at Montreal, Montreal H3C 3P8, Quebec, Canada.

[16] State Key Laboratory of Soil Erosion and Dryland Farming on the Loess Plateau, College of Forestry, Northwest A & F University, Yangling 712100, China.

[17] Environmental Sciences Division and Climate Change Science Institute, Oak Ridge National Laboratory, Oak Ridge, Tennessee 37831-6301, USA.

[18] International Center for Climate and Global Change Research, School of Forestry and Wildlife Sciences, Auburn University, Auburn, AL 36849, USA.

[19] Institute of Arctic Biology, University of Alaska Fairbanks, Fairbanks, Alaska 99775, USA.

[20] Max Planck Institute for Meteorology, 20146 Hamburg, Germany.

[21] School of Geosciences, University of Edinburgh, Edinburgh EH9 3FF, UK.

[22] Research School of Biology, Australian National University, Canberra, ACT 2601, Australia.

[23] Norwegian Institute of Bioeconomy Research, 1431 Ås, Norway.

[24] Woods Hole Research Center, Falmouth, Massachusetts 02540-1644, USA.

[25] College of Life and Environmental Sciences, University of Exeter, Exeter EX4 4RJ, UK.

[26] CSIRO Oceans and Atmosphere, PMB #1, Aspendale, Victoria 3195, Australia.

[27] Center for Ecosystem Sciences and Society, Department of Biological Sciences, Northern Arizona University, Flagstaff, AZ 86011, USA.

*Correspondence to*: Donghai Wu (donghai.wu@pku.edu.cn)

**Abstract**

Changes in precipitation variability are known to influence grassland growth. Field measurements of aboveground net primary productivity (ANPP) in temperate grasslands suggest that both positive and negative asymmetric responses to changes in precipitation may occur. Under normally variable precipitation regimes, wet years typically result in ANPP gains being larger than ANPP declines in dry years (positive asymmetry), whereas increases in ANPP are lower in magnitude in extreme wet years compared to reductions during extreme drought (negative asymmetry). Whether ecosystem models that couple carbon-water system in grasslands are capable of simulating these non-symmetrical ANPP responses is an unresolved question. In this study, we evaluated the simulated responses of temperate grassland primary productivity to scenarios of altered precipitation with fourteen ecosystem models at three sites, Shortgrass Steppe (SGS), Konza Prairie (KNZ) and Stubai Valley meadow (STU), spanning a rainfall gradient from dry to moist. We found that: (1) Gross primary productivity (GPP), NPP, ANPP and belowground NPP (BNPP) showed concave-down nonlinear response curves to altered precipitation in all the models, but with different curvatures and mean values. (2) The slopes of spatial relationships (across sites) between modeled primary productivity and precipitation were steeper than the temporal slopes obtained from inter-annual variations, consistent with empirical data. (3) The asymmetry of the responses of modeled primary productivity under normal inter-annual precipitation variability differed among models, and the median of the model-ensemble suggested a negative asymmetry across the three sites, in contrast to empirical studies. (4) The median sensitivity of modeled productivity to rainfall consistently suggested greater negative impacts with reduced precipitation than positive effects with increased precipitation under extreme conditions. This study indicates that most models overestimate the extent of negative drought effects and/or underestimate the impacts of increased precipitation on primary productivity under normal climate conditions, highlighting the need for improving eco-hydrological processes in models.

## 1 Introduction

Precipitation is a key climatic determinant of ecosystem productivity, especially in grasslands which limits productivity over the majority of the globe (Lambers et al., 2008; Sala et al., 1988; Hsu et al., 2012; Beer et al., 2010). Climate models project substantial changes in amounts and frequencies of precipitation regimes worldwide, and this is supported by observational data (Karl and Trenberth, 2003; Donat et al., 2016; Fischer and Knutti, 2016). Potential for increasing occurrence and severity of droughts and increased heavy rainfall events related to global warming will likely affect grassland growth (Knapp et al., 2008; Gherardi and Sala, 2015; Lau et al., 2013; Reichstein et al., 2013). As a consequence, better understanding of the responses of grassland productivity to altered precipitation is needed to project future climate-carbon interactions, changes in

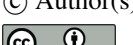



ecosystem states, and to gain better insights on the role of grasslands, in supporting crucial ecosystem services (e.g. livestock production).

Gross primary productivity (GPP) of ecosystems is controlled by environmental conditions, in particular water availability (Jung et al., 2017), and by biotic factors affecting leaf photosynthetic rates and stomatal conductance, which scale up to canopy-

level functioning (Chapin III et al., 2011). About half of GPP is respired while the remainder, net primary productivity (NPP), is primarily invested in plant biomass production, including photosynthetic and structural pools aboveground (foliage and stem) and belowground (roots) (Waring et al., 1998; Chapin III et al., 2011). NPP responses to precipitation have been observed using multi-year, multi-site observations (Hsu et al., 2012; Estiarte et al., 2016; Knapp and Smith, 2001; Wilcox et al., 2015). Positive empirical relationships between grassland aboveground NPP (ANPP) and precipitation (P) have been found in spatial

gradients across sites (Sala et al., 1988) and from temporal variability at individual sites (Huxman et al., 2004; Knapp and Smith, 2001; Roy et al., 2001; Hsu et al., 2012). The P-ANPP sensitivities obtained from spatial relationships are usually higher than those obtained by temporal relationships (Estiarte et al., 2016; Fatichi and Ivanov, 2014; Sala et al., 2012). Possible mechanisms behind the steeper spatial relationship may be (1) a 'vegetation constraint' reflecting the adaptation of plant communities over long time scales in such a way that grasslands make the best use of the water received from rainfall for

growth, and (2) the spatial variation in structural and functional traits of ecosystems (soil properties, nutrient pools, plant and microbial community composition) that constrain local P-ANPP sensitivities (Lauenroth and Sala, 1992; Smith et al., 2009; Wilcox et al., 2016). For projecting the effect of climate change on grassland productivity in the coming decades, inter-annual relationships are arguably more informative than spatial relationships because spatial relationships reflect long-term adaptation of ecosystems, and because P-ANPP relationships from spatial gradients are confounded by the co-variation of gradients in

other environmental variables (e.g. temperature and radiation) and soil properties (Estiarte et al., 2016; Knapp et al., 2017b).

In temporal P-ANPP relationships, an important observation is the asymmetric responses of productivity in grasslands to altered precipitation (Knapp et al., 2017b; Wilcox et al., 2017). Compared to negative anomalies of ANPP from years with decreased precipitation, positive anomalies of ANPP during years with increased precipitation were usually found to have a larger absolute magnitude, suggesting a convex positive response (positive asymmetry) (Bai et al., 2008; Knapp and Smith,

2001; Yang et al., 2008). Yet, when grasslands are subject to extreme precipitation anomalies that fall beyond the range of normal inter-annual variability, an extreme dry year is associated with a larger absolute ANPP loss than the gain found during an extreme wet year. This suggests a convex negative response (negative asymmetry) when considering a larger range of rainfall anomalies than the current inter-annual regime (Knapp et al., 2017b). This is also supported by current dynamical global vegetation models, which suggest a stronger response to extreme dry conditions compared to extreme wet conditions

(Zscheischler et al., 2014). The sign of the asymmetric response of grassland productivity to altered rainfall thus depends on the magnitude of rainfall anomalies, the size-distribution of rainfall events, and ecosystem mean state (Gherardi and Sala, 2015;

Hoover and Rogers, 2016; Parolari et al., 2015; Peng et al., 2013).

Relationships between precipitation and grassland productivity have previously been studied with site observations (Hsu et al., 2012; Knapp et al., 2017b; Luo et al., 2017; Wilcox et al., 2017; Estiarte et al., 2016), but they remain to be quantified and characterized in ecosystem models used for diagnostic and future projections of the coupled carbon-water system in grasslands,

in particular grid-based models used as the land surface component of Earth System Models. In this study, we aim to evaluate the responses of simulated productivity to altered precipitation from fourteen ecosystem models at three sites representing dry, mesic, and moist rainfall regimes. The specific objectives of this study are to: (1) analyze the response of simulated productivity fluxes (GPP, NPP, ANPP and BNPP) for a large range of altered precipitation amounts across the three sites, and test if the P-productivity sensitivities become weaker at moister sites; (2) test if the P-productivity sensitivities of spatial relationships are

greater than the temporal ones in the models like found in the observations; (3) test if models reproduce the observed asymmetric responses under inter-annual precipitation conditions; (4) assess the P-productivity sensitivities related to different precipitation regimes including normal and extreme conditions, and to test in particular if sensitivities for extreme drought conditions are stronger than those for high-rainfall conditions. In addition, collaboration between a large group of modelers and site investigators to highlight major problems in current ecosystem models also set this work as an introduction for future

studies on model-experiment combination. As far as we know, this is the first model-experiment interaction study elucidating the precipitation-productivity relationships in multiple sites.

## 2    Materials and methods

### 2.1 Experimental sites

We conducted model simulations using three sites: the Shortgrass Steppe (SGS) site at the Central Plains Experimental Range,

the Konza Prairie Biological Station (KNZ) site, and the Stubai Valley meadow (STU) site. These sites represent three grassland types spanning a productivity gradient from dry to moist climatic conditions. The dry SGS site is located in northern Colorado, USA (Knapp et al., 2015; Wilcox et al., 2015). The KNZ site is a native $C_4$-dominated mesic tallgrass prairie in the Flint Hills of northeastern Kansas, USA (Heisler-White et al., 2009; Hoover et al., 2014). The moist site of STU is a subalpine meadow located in Austria Central Alps near the village of Neustift (Bahn et al., 2006; Bahn et al., 2008; Schmitt et al., 2010).

Experimental measurements of annual ANPP were carried out spanning different time ranges. Estimated mean ANPP for SGS, KNZ and STU sites are 91 g DM (dry mass) m$^{-2}$ yr$^{-1}$, 387 g DM m$^{-2}$ yr$^{-1}$, and 525 g DM m$^{-2}$ yr$^{-1}$. Details of the ecological and environmental factors are summarized in Table 1.

### 2.2 Ecosystem model simulations

In order to test the hypothesis of an asymmetric response of productivity to variable rainfall (Knapp et al., 2017b), simulations

were conducted for fourteen ecosystem models CABLE, CLM45-ORNL, DLEM, DOS-TEM, JSBACH, JULES, LPJ-GUESS, LPJmL-V3.5, ORCHIDEE-2, ORCHIDEE-11, T&C, TECO, TRIPLEX-GHG and VISIT all using the same protocol defined by the precipitation subgroup of the model-experiment interaction study (Table 2). At all three grassland sites, observed and altered multi-annual hourly rainfall forcing time series were combined with observations of other climate variables. These

variables were air temperature, incoming solar radiation, air humidity, wind speed and surface pressure. Simulated productivity during the observational period is influenced at least in some models (for instance those having C-N interactions) by historical climate change and $CO_2$ changes since the pre-industrial period. Thus instead of assuming that productivity was in equilibrium with current climate, historical reconstructions of meteorological variables from gridded CRUNCEP data at 1/2 hourly time step (Wei et al., 2014) were combined and bias-corrected with site observations to provide bias corrected historical forcing time series from 1901 to 2013 (CRUNCEP-BC). In addition to the observed current climate defining the ambient simulation,

nine altered rainfall forcing datasets were constructed by decreasing/increasing the amount of precipitation in each precipitation event by -80%, -70%, -60%, -50%, -20%, +20%, +50%, +100% and +200% during the time-span of productivity observations at each site, leaving all other meteorological variables unchanged and equal to the observed values. Modelers performed all simulations described below based on the same protocol (see below) and the model output was compared with

measured ecosystem productivities (GPP, NPP, ANPP and BNPP).

- Simulation S0 spin-up: models simulated an initial steady state spin-up run for water and biomass pools under pre-industrial conditions using the 1901-1910 CRUNCEP-BC climate forcing in a loop and applying fixed atmospheric $CO_2$ concentration at the 1850 level.

- Simulation S1 historical simulation from 1850 until the first year of measurement (1986 for SGS, 1982 for KNZ, and 2009

for STU): starting from the spin-up state, models were prescribed with increasing atmospheric $CO_2$ concentrations and dynamic historical climate from CRUNCEP-BC. Because there is no CRUNCEP-BC data for 1850-1900, the CRUNCEP-BC climate data from 1901 to 1910 was repeated in a loop instead.

- Simulation SC1 ambient simulation for the measurement periods (1986-2009 for SGS, 1982-2012 for KNZ, and 2009-2013 for STU) with observed $CO_2$ concentrations and meteorological data corresponding to gap-filled site observations

at the hourly or half-hourly scale.

- Simulations SP1-SP9 altered precipitation simulations for the measurement periods (1986-2009 for SGS, 1982-2012 for KNZ, and 2009-2013 for STU), starting from the initial state in the start year of the period and run using the nine altered rainfall forcing datasets with observed $CO_2$ concentration.



### 2.3 Metrics of the response of productivity to precipitation changes

We calculated three different indices to analyze the nonlinearity of modeled and observed - when data were available - response of productivity to precipitation: (1) the parameters of the curvilinear P-productivity relationships across the full range of altered precipitation, based on fits to model output for the ambient (SC1) and altered (SP) simulations; (2) the asymmetry of P-productivity for current inter-annual variability, based on SC1 and was also calculated with observations for ANPP; and (3) the sensitivity of productivity to P for altered versus ambient simulations, based on SC1 and SP results. Methods for the three indices are introduced in the following.

#### 2.3.1 Curvilinear P-productivity relationships across the entire range of altered P

In general, plant productivity increases with increasing precipitation, and saturates when photosynthesis becomes less limited by water scarcity. We fitted the response of simulated productivity to altered precipitation using the Eq. (1):

$$y = a(1 - e^{-bx}) \tag{1}$$

Where the independent variable $x$ is the mean annual precipitation (mm), and the dependent variable $y$ one of the productivities (GPP, NPP, ANPP and BNPP). Parameter $a$ (g C m$^{-2}$ yr$^{-1}$) is the maximum value of productivity at high precipitation; and parameter $b$ (mm$^{-1}$) is the curvature of modeled productivity to altered precipitation.

#### 2.3.2 Asymmetry index from inter-annual productivity and precipitation

In order to characterize the asymmetry of productivity to precipitation, we define the asymmetry index (AI) from inter-annual productivity and precipitation data as follows:

$$AI = R_p - R_d \tag{2}$$

where $R_p$ is the relative productivity pulse in wet years, and $R_d$ is the relative productivity decline in dry years defined by:

$$R_p = (med(f_{p90}) - \bar{f})/\bar{f} \tag{3}$$

$$R_d = (\bar{f} - med(f_{p10}))/\bar{f} \tag{4}$$

where $f$ is the inter-annual productivity, being a function of environmental factors from models or observation; $\bar{f}$ is mean annual productivity in the period of measurements (Table 1); $med(f_{p90})$ is the median value of productivities in wet years with annual precipitation higher than the 90$^{th}$ percentile level; $med(f_{p10})$ is median value of productivities in all the dry years when annual precipitation is lower than the 10$^{th}$ percentile level.

#### 2.3.3 Sensitivity of productivity to altered versus inter-annual precipitation variability

For altered precipitation, in particular for the extreme SP simulations where mean precipitation was altered and individual





years went beyond the normal range of inter-annual variability, we wanted to test the hypothesis whether the asymmetry response becomes negative, that is the impacts of extreme dry conditions on productivity are much greater than the positive effects of extreme wet scenarios (Knapp et al., 2017b). We defined the sensitivity of productivity to altered rainfall conditions ($S$) by:

$$S = (\overline{f_{P_a}} - \overline{f_{P_c}})/(|\overline{P}_a - \overline{P}_c|) \tag{5}$$

where $\overline{f_{P_a}}$ and $\overline{f_{P_c}}$ are the mean productivities of altered and ambient simulations; $\overline{P}_a$ and $\overline{P}_c$ are the mean annual precipitation amounts in altered and ambient simulations.

## 3    Results

### 3.1 Inter-model differences in the ambient simulation

Estimates of GPP for the ambient simulation varied by a factor of ~3.3 between models at STU, ranging from 639 g C m$^{-2}$ yr$^{-1}$ (VISIT) to 2104 g C m$^{-2}$ yr$^{-1}$ (CABLE) (Fig. 1a), by a factor of ~7.0 from 269 g C m$^{-2}$ yr$^{-1}$ (CLM45-ORNL) to 1892 g C m$^{-2}$ yr$^{-1}$ (CABLE) at KNZ (Fig. 1d) and by a factor of ~5.5 from 197 g C m$^{-2}$ yr$^{-1}$ (DLEM) to 1088 g C m$^{-2}$ yr$^{-1}$ (CABLE) at SGS (Fig. 1g). Similarly, large variations across models were found from the control NPP simulations at the three sites (Fig. 1b, e, h). The carbon use efficiency (CUE), defined from the ratio of NPP and GPP, showed large differences across models, and ranged from 0.44 to 0.66 at STU, from 0.27 to 0.62 at KNZ and from 0.42 to 0.67 at SGS for the ambient simulation (Fig. 1c, f, i).

### 3.2 Responses of productivity to altered precipitation

At SGS and KNZ, simulated GPP and NPP increased with increasing precipitation. In contrast, at the moist STU, most models showed saturation in productivity for precipitation above ambient values (Fig. 1). Along with increasing precipitation, GPP and NPP showed nonlinear concave-down response curves in all models, with different curvatures $b$ and maximum productivity $a$ (Fig. S1). The median values of the curvature parameter $b$ fitted from Eq. (1) to each modeled GPP across the full range of altered precipitation are $4.1*10^{-3}$ mm$^{-1}$ at STU, $1.8*10^{-3}$ mm$^{-1}$ at KNZ and $1.7*10^{-3}$ mm$^{-1}$ at SGS (Fig. S1); Here lower $b$ values indicate a flatter curvature. The steeper curvature at STU despite saturation above ambient precipitation indicates a steeper decline of productivity for precipitation set below ambient for this site compared to KNZ and SGS (Fig. 1). Additionally, the ranking of models for $b$ and $a$ differed between the three sites (Fig. S1).

The responses of GPP and NPP to altered precipitation were proportional to each other for each model, and as a result changes in CUE were very small compared to the background CUE differences diagnosed in the ambient simulation (Fig. 1c, f, i). However, JSBACH and LPJmL-V3.5 produced a sharp decline of CUE below ambient precipitation at SGS and KNZ.





Only seven models simulated ANPP and BNPP separately (Fig. 2). The responses of ANPP and BNPP to altered precipitation were similar to those of GPP and NPP. When fitting Eq. (1) to P-ANPP (Fig. S2), the curvatures $b$ ranged from $3.0*10^{-3}$ mm$^{-1}$ (ORCHIDEE-11) to $9.2*10^{-3}$ mm$^{-1}$ (TECO) at STU, from $0.7*10^{-3}$ mm$^{-1}$ (TRIPLEX-GHG) to $6.1*10^{-3}$ mm$^{-1}$ (VISIT) at KNZ, and from $0.9*10^{-3}$ mm$^{-1}$ (T&C) to $2.3*10^{-3}$ mm$^{-1}$ (CLM45-ORNL) at SGS; the modeled maximum values $a$ for ANPP ranged

between 173 g C m$^{-2}$ yr$^{-1}$ (VISIT) and 827 g C m$^{-2}$ yr$^{-1}$ (TECO) at STU, 49 g C m$^{-2}$ yr$^{-1}$ (CLM45-ORNL) and 557 g C m$^{-2}$ yr$^{-1}$ (ORCHIDEE-2) at KNZ, and 94 g C m$^{-2}$ yr$^{-1}$ (CLM45-ORNL) and 523 g C m$^{-2}$ yr$^{-1}$ (ORCHIDEE-2) at SGS.

The ANPP:NPP ratio, i.e., aboveground carbon allocation, showed a nonlinear increase (concave-down) with increasing precipitation in ORCHIDEE-2 and ORCHIDEE-11, a nonlinear decrease (concave-up) in T&C due to translocation of C reserves from roots and only minor changes in other models (Fig. 2c, f, i).

**3.3 Temporal versus spatial slopes of P-productivity**

From the ambient simulations, ensemble model results indicate that the mean slopes of the spatial relationships were steeper than the temporal slopes for GPP, NPP and ANPP for the subset of models that simulated this flux, while these differences in slopes were less obvious for BNPP (Fig. 3). For P-ANPP temporal slopes of the ambient simulation across the three sites, we compared model results with site-observations (Fig. 3c). Observed and modeled temporal slopes decreased from dry (SGS) to

moist (STU) site, from 0.10 g C m$^{-2}$ mm$^{-1}$ to 0.05 g C m$^{-2}$ mm$^{-1}$ in the observations, and from 0.14±0.16 g C m$^{-2}$ mm$^{-1}$ to 0.03±0.15 g C m$^{-2}$ mm$^{-1}$ for the model ensemble mean. Although there were some discrepancies in the range of spatial and temporal slopes across models (Fig. S3), the multi-model ensemble mean captured the key observation of steeper spatial than temporal slopes for ANPP (Fig. 3).

**3.4 Asymmetry of the inter-annual primary productivity response to precipitation**

The asymmetry of each model was diagnosed using the asymmetry index (AI; Eq. (2)), which showed large variation across models (Fig. 4, S4). Considering all the models as independent samples, the median AI of GPP and NPP showed significantly negative values at $p < 0.1$ level for SGS (median value of -0.12±0.11 and -0.24±0.11 respectively). Hence, for SGS simulated declines of GPP and NPP in dry years were proportionately larger than the increases in wet years. For STU, the AI values were only slightly negative (median for GPP -0.03±0.03 and for NPP -0.04±0.03), while AI was very close to zero at KNZ. By

contrast, observation-based AI values, estimated from long-term inter-annual ANPP measurements, suggest a decrease from positive (0.32 for SGS and 0.20 for KNZ) to negative (-0.21 for STU). At the dry (SGS) and mesic (KNZ) sites (Fig. S4), most of model simulations overestimated the extent of negative drought effects in dry years ($R_d$) and/or underestimated the positive impacts on ANPP in wet years ($R_p$). At the moist site (STU), models agreed with observations regarding the negative sign of AI (negative asymmetry) but AI magnitude is not well captured.



### 3.5 Sensitivities of primary productivity to altered precipitation

The model-derived sensitivities given by Eq. (5) generally indicated greater negative impacts of reduced precipitation than positive effects of increased precipitation under both normal (inter-annual) and extreme conditions (Fig. 5).

Primary productivity at the dry site (SGS) was more sensitive to precipitation changes compared to the moist site (STU). Along

with increases in precipitation, the largest sensitivity values were found for SGS ($1.35\pm0.63$ g C m$^{-2}$ mm$^{-1}$ for GPP, $0.68\pm0.32$ g C m$^{-2}$ mm$^{-1}$ for NPP, $0.24\pm0.13$ g C m$^{-2}$ mm$^{-1}$ for ANPP and $0.16\pm0.02$ g C m$^{-2}$ mm$^{-1}$ for BNPP) and then KNZ ($0.32\pm0.49$ g C m$^{-2}$ mm$^{-1}$ for GPP, $0.20\pm0.22$ g C m$^{-2}$ mm$^{-1}$ for NPP, $0.13\pm0.08$ g C m$^{-2}$ mm$^{-1}$ ANPP and $0.06\pm0.04$ g C m$^{-2}$ mm$^{-1}$ for BNPP) when precipitation was altered by +20%. The values of S decreased with further increased precipitation, indicating that additional water does not increase productivity in the same proportion exceeding a certain threshold. In contrast to SGS, the

values of sensitivity for both GPP and NPP at STU are close to zero in response to added precipitation conditions, implying that the precipitation above ambient was not a limiting factor for grassland production in the models at this site.

The values of sensitivity decreased with reduced precipitation at KNZ and SGS, indicating larger negative impacts on primary productivity when conditions become drier. For the moist site of STU, primary productivities showed less sensitivity to moderately (up to -50%), and sensitivity only increased with more extreme rainfall alterations out of $3\sigma$ (~50% precipitation

change). Additionally, the values of S for ANPP were smaller than those of BNPP at KNZ and SGS, while there were no differences between ANPP and BNPP at STU (Fig. 5). Thus, models suggest that the dry site (SGS) was particularly vulnerable to climatic changes, caused through already slight alterations in precipitation (increase or decrease), while moist site (STU) was more robust in response to altered rainfall.

### 4    Discussion

#### 4.1 Modeled responses of productivities to altered precipitation

Only a few previous model studies have directly linked the responses of grassland primary productivity to altered precipitation at field observation sites (Peng et al., 2013; Zhou et al., 2008; Fatichi and Ivanov, 2014). Peng *et al.* (2013) conducted an analysis with a process-based model (ORCHIDEE 2-layer version) to address how precipitation changes regulate carbon cycling in a semi-arid grassland ecosystem in northern China. Zhou *et al.* (2008) modeled the patterns of nonlinearity in

ecosystem responses to multiple climatic factors (temperature, precipitation and $CO_2$) at a tallgrass prairie site in the central United States with the TECO model, which is included in this study. Their main results suggested similar nonlinear response curves showing negative asymmetric responses of greater NPP losses to decreased precipitation relative to NPP gains in response to increased precipitation (Fig. 1, 2). However, these studies did not compare effects across precipitation gradients and across multiple ecosystem models.



In general, precipitation in ecosystem models is distributed through three pathways (Smith et al., 2014b): (1) intercepted by vegetation and subsequently evaporated or falling on the ground; (2) infiltrated into the upper soil layers with subsequent evaporation, root water uptake and plant transpiration, or percolated down to deeper layers to form ground water; (3) runoff from the soil surface if the intensity of precipitation exceeds infiltration rates. In reality as well as in models, soil moisture

rather than precipitation is the variable regulating vegetation growth, and biological responses to changes in precipitation are manifested as functions of soil moisture in different soil layers (Sitch et al., 2003; Smith et al., 2014b; Vicca et al., 2012). We calculated the surface soil water content (SSWC, 0-20cm depth converted from reported soil layers) and total soil water content (TSWC) under ambient and altered precipitation as simulated by the fourteen models, and we found different patterns with parabolic, asymptotic and threshold-like nonlinear curves, which is similar to the response curves of primary productivity at

the three sites (Fig. S5, S6). For the moist STU, SSWC and TWSC did not show obvious changes in response to increased precipitation since soil moisture at this site is often relatively near field capacity, while the SSWC and TSWC quickly decreased with decreasing in precipitation (Fig. S5, S6). In contrast, SSWC and TSWC at SGS showed significant increases in response to altered increased precipitation, and slow decreases for decreased precipitation, because the soil was already very dry under average ambient conditions. Thus, changes of SWC in response to precipitation contribute to driving the different response

patterns of simulated primary productivity across the grassland sites.

The responses of primary productivity to precipitation in models might also be driven by the intrinsic structure and parameterizations of vegetation functioning besides changes of soil moisture (Gerten et al., 2008), which account for the large spread in the values of *b* and *a* among models at the three sites (Figure 1, 2, S1, S2). For example, carbon-nitrogen cycle coupling in ecosystem models reduced the simulated vegetation productivity relative to a carbon-only counterpart model

(Thornton et al., 2007; Zaehle et al., 2010). Of those models used in this study, only five of the 14 models include carbon-nitrogen-water interactions (Table 2, S1, S2). We calculated the model ensemble of productivity for this group of carbon-nitrogen models and carbon-only models across altered and ambient precipitation simulations at the three sites, and then fitted the P-productivity responses with Eq. (1) (Fig. S7, S8, S9). We found that carbon-nitrogen models generally produce a weaker GPP, NPP and ANPP response to precipitation than carbon-only models, and similar responses for BNPP. The latter may be

explained by fixed root profiles in most models (Table S13). Our findings suggest that N interactions in ecosystem models reduced the P-productivity sensitivities, but should be confirmed using the same model prescribed with different N availability. In addition to the influence of nutrient cycling, different definitions of vegetation compositions (C3/C4) (Table S14), root profiles (Table S13), phenology (Table S9) and carbon allocation (Table S4) at the three sites may also contribute to the large variations of modeled P-productivity responses and demands for more accurate calibration of models to the specificity of the

local sites in future model intercomparison studies.





### 4.2 Comparison of modeled and observed responses of productivity to altered precipitation

Steeper spatial than temporal slopes of ANPP to precipitation are usually explained by two hypotheses: (1) 'vegetation constraint' effects on ANPP responses to precipitation play a more important role in the temporal as compared to the spatial domain (Knapp et al., 2017b; Estiarte et al., 2016); (2) biogeochemistry (mainly resource limitations) and confounding factors

(e.g. temperature and radiation), rather than species attributes, constrain community level ANPP in response to precipitation (Huxman et al., 2004). Thus, the former theory stresses more long-term intrinsic ecosystem properties, while the latter one underlines the effects of external environmental factors. The current models tested here captured the relative magnitude of the difference between temporal and spatial slopes (Fig. 3c), which suggested that the models adequately considered the key processes underlying carbon-water interactions across different grassland sites. Only few grassland experiments have assessed

BNPP (Luo et al., 2017), leaving the question open whether the minor differences between temporal and spatial slopes for BNPP responses to precipitation as simulated by the models, correspond to experimental observations (Fig. 3d).

The asymmetry index obtained from available long-term ANPP and precipitation observations reported positive values at SGS and KNZ (Fig. 4c), which suggested greater declines of ANPP in dry years than increases in wet years (Knapp and Smith, 2001). Knapp *et al* (2017b) proposed the following underlying mechanisms: (1) In dry years, the carry-over effects of soil

moisture from previous years alleviate strong declines of ANPP (Sala et al., 2012), which is usually treated as a time-lag effect (Wu et al., 2015). Additionally, rain use efficiency also increases with water scarcity, meaning that less water is lost through runoff (Gutschick and BassiriRad, 2003; Huxman et al., 2004). (2) In wet years, other resources like nutrient availability, may increase with increasing precipitation, contributing to a supplementary increase of ANPP (Knapp et al., 2017b; Seastedt and Knapp, 1993). In contrast, the negative asymmetry index derived from observations at the moist STU suggest that this process

is not dominant for this site, while temperature and/or light limitations that are associated with rainy periods may become important during wet years and neutralize the effect of increased precipitation on ANPP (Fig. S4) (Nemani et al., 2003; Wu et al., 2015; Wohlfahrt et al., 2008).

In our results, most models did not capture the sign of observed asymmetry indices across the three sites (Fig. 4c), which suggests that some of the underlying processes (combined carbon-nutrient interactions, time-lag effects, dynamic root growth

allowing variation in accessible soil water) are not accurately implemented in the models. For example, grassland root depth affects ecosystem resilience to environmental stress such as drought, and arid and semi-arid grasses that have extensive lateral roots or possibly deep roots show relatively strong resistance (Fan et al., 2017). However, most models currently consider only two types of grasslands, C3 and C4 (Table S14) with fixed root profiles along with prescribed soil layers (Table S13). This is potentially unrealistic for semi-arid grass roots and can lead to models underestimating the accessible water and the resistance

to drought. The latter is a key candidate especially for explaining the negative asymmetry index at dry SGS.



The model results for ANPP ambient simulations generally suggest negative asymmetric responses for normal precipitation variability at dry (SGS) and mesic (KNZ) sites (Fig. 5c). This contrasts with a meta-analysis of grassland precipitation manipulation experiments (Wilcox et al., 2017) and with the P-ANPP conceptual model (Knapp et al., 2017b), which suggest a positive asymmetry response in the range of normal rainfall variation. This emphasizes the finding that most models overestimate drought effects and/or underestimate wet year impacts on primary productivity of dry and mesic sites for current precipitation variability. When moved to extreme conditions with modified precipitation, models were in line with the hypothesis and the data showing that ANPP saturates in very wet conditions but declines strongly in very dry conditions (Knapp et al., 2017b). Because there are still only few extreme precipitation manipulation experiments (Knapp et al., 2017a) and limited associated understanding and model development (Meir et al., 2015), we are currently unable to evaluate the point of ecosystem collapse and point of release from water limitations. For BNPP sensitivities to altered precipitation, meta-analysis of previous experiments indicated symmetric responses to increasing and decreasing rainfall (Luo et al., 2017; Wilcox et al., 2017), which may be regulated by allocation controls on the ratio of ANPP and BNPP to total NPP in response to altered precipitation. However, in the participating models, BNPP shows a negative asymmetric responses to altered rainfall (Fig. 5d), which may reflect a shortcoming of carbon-water interactions in the belowground ecosystems.

## 4.3 Uncertainties, knowledge gaps and suggestions of further work

In this work, we applied two methods to characterize the asymmetry responses in normal precipitation range using inter-annual variability of present conditions and forcing models with continuously modified precipitation amounts. Asymmetry indices from the inter-annual gross and net primary productivities suggest large uncertainties (Fig. 4), while the sensitivity analysis to changes in mean precipitation reported clear responses (Fig. 5). This can be explained by the conditions of other climatic factors (for example, temperature, radiation, and vapor pressure) that may be different in corresponding dry and wet years, and discrepancies of precipitation timing and frequency also exist between dry and wet years. All these uncontrolled factors may contribute to the large uncertainties of asymmetric responses from inter-annual variations (Chou et al., 2008; Peng et al., 2013; Robertson et al., 2009).

Although the carbon-water interactions in current models have been improved during the last decades, there still exist large gaps for accurately diagnosing the inadequate representation of processes and parameterizations. Gaps that should be considered in future work include: (1) Soil moisture directly affects the growth of grassland, and the sensitivity of primary productivity to SWC is a cornerstone variable in models (Smith et al., 2014b). We recommend that models simulate SWC in the same soil layer as experiments in following studies, also considering the local soil textures. This will help in figuring out the bias of modeled sensitivities to precipitation and to check explicitly the sensitivity of vegetation productivity to change in SWC; (2) Responses of ANPP and BNPP to altered precipitation are different, however, there are still only few control

experiments for BNPP to be used for evaluating the corresponding processes in models (Luo et al., 2017; Wilcox et al., 2017); (3) Consideration of other processes such as responses of primary productivity to irrigation or drought manipulations, and time-lag effects of droughts (Hoover and Rogers, 2016; Hoover et al., 2014; Sala et al., 2012; Wang et al., 2014). To study such effects, modelers will need to simulate the control experiments corresponding to the real local manipulations applied by

field scientists, e.g., considering vegetation composition, root profiles, nutrient cycling, phenology and carbon allocation as close as possible to local conditions. This should be a priority for future model-experiment interaction studies.

## 5    Conclusions

Regardless of limitations related to the idealized nature of the precipitation manipulation scenarios in this study, this is the first study where a large group of modelers simulated the response of grassland primary productivity to precipitation using long-

term observations for evaluating the asymmetry responses to altered precipitation. Results indicate that most models do not capture the observed asymmetry responses in the normal precipitation range, suggesting an overemphasis of the drought effects and/or underestimating the watering impacts on primary productivity in the normal state, which may be the result of inadequate representation of key processes (e.g. carbon-nitrogen coupling and carbon-water coupling) and parameterizations (e.g. vegetation composition and root profile). This study paves the path for further analyses in this direction, and follow-up model-

experiment interaction studies. Collaboration between modelers and site investigators needs to be strengthened to make use of a more data beyond ANPP and improve specific processes in ecosystem models. This will eventually allow us to produce more reliable carbon-climate projections when facing climate extremes in the future.

**Data availability**

All the modeled outputs in the first model-experiment interaction study are available via contacting Donghai Wu

(donghai.wu@pku.edu.cn).

**Competing interests**

The authors declare that they have no conflict of interest.

**Acknowledgements**

This study was supported by National Natural Science Foundation of China (41530528). We also acknowledge support from

the ClimMani COST action (ES1308). S.V. is a postdoctoral fellow of the Fund for Scientific Research - Flanders. M.K. acknowledges support from the EU FP7 project LUC4C, grant 603542. We thank Jeffrey S. Dukes, Shiqiang Wan and the




organizers of the conference for model-experiment interaction study in Beijing. We thank Sibyll Schaphoff, Werner von Bloh, Susanne Rolinski and Kirsten Thonicke from PIK and Matthias Forkel from TU Vienna for their support of the LPJmL code.

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

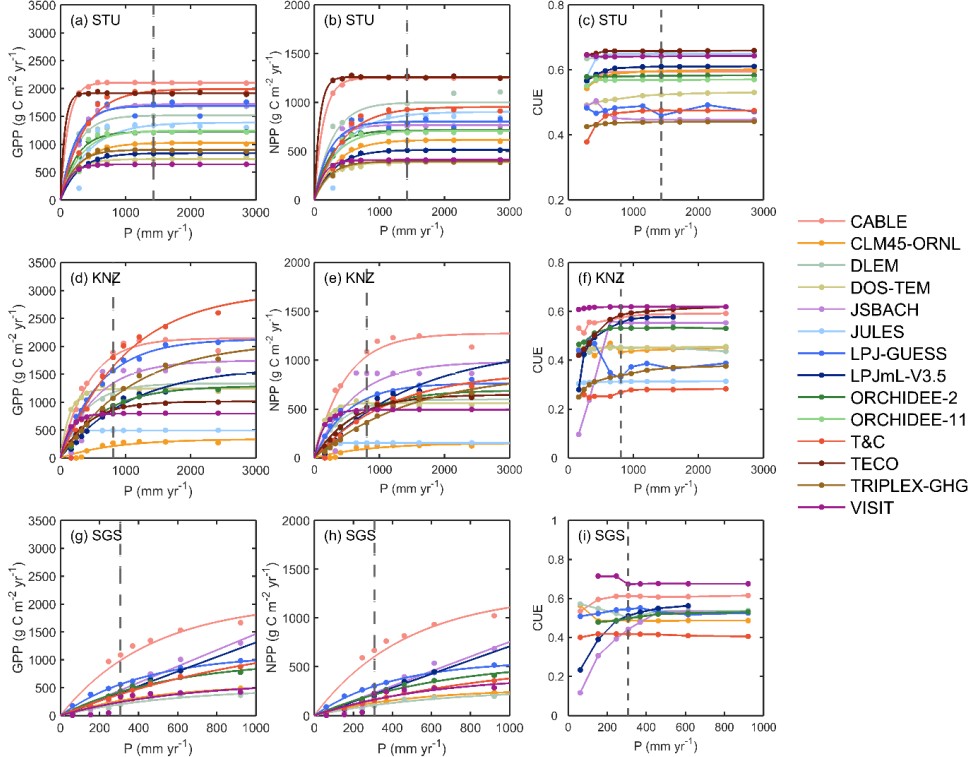

**Figure 1** Responses of simulated annual GPP (left column), NPP (central column) and CUE (NPP / GPP; right column) to

altered and ambient precipitation (P) levels at the three sites STU, KNZ and SGS. The fitted equation is Eq. (1) for GPP and

NPP (see Fig. S1 for fitted *a* and *b*). The grey dashed line represents ambient precipitation.





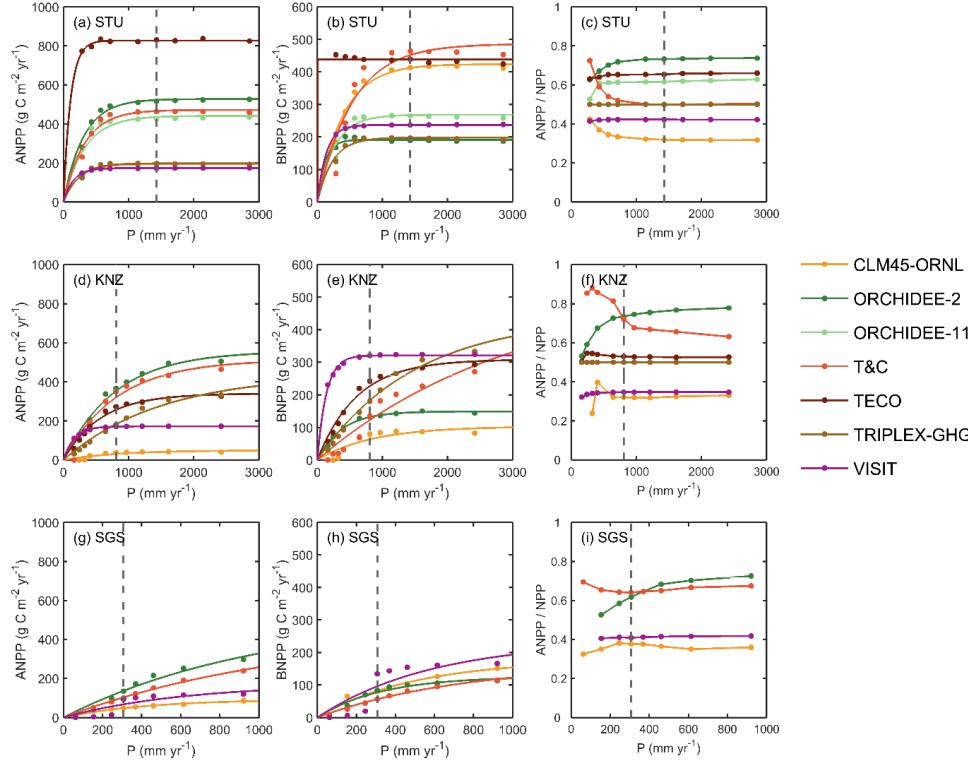

**Figure 2** Responses of simulated annual ANPP (left column), BNPP (central column) and the ratio of ANPP and NPP (right column) to altered and ambient precipitation (P) levels at the three sites STU, KNZ and SGS. The fitted equation is Eq. (1) for ANPP and BNPP (see Fig. S2 for fitted *a* and *b*). The grey dashed line represents ambient precipitation.

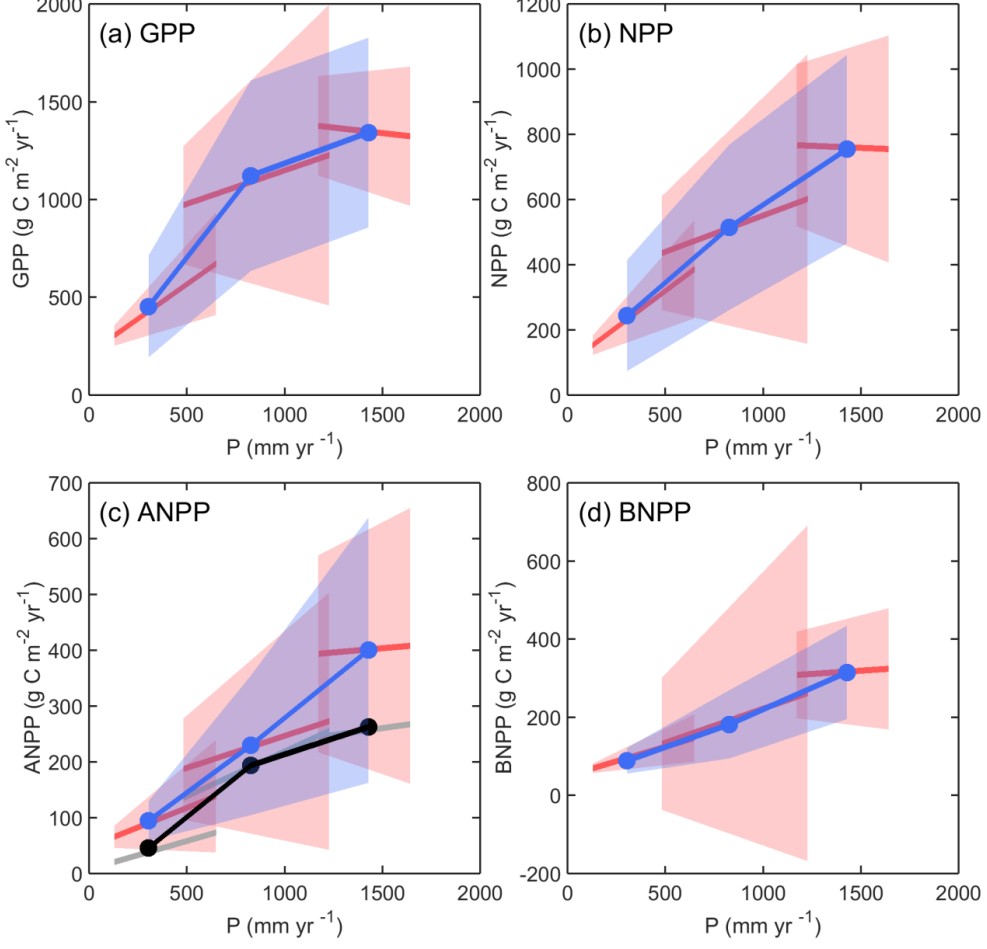

**Figure 3** Relationships between GPP (a), NPP (b), ANPP (c) and BNPP (d) and precipitation (P) derived from multi-year

ambient simulations (SC1) in two ways. Temporal slopes are site based and relate inter-annual variability in P to inter-annual

variability in the productivities. Spatial slopes relate mean annual P to mean annual productivity across three sites. In each

5    panel, SGS, KNZ and STU are from dry to moist, given from left to right. The red lines are the mean of modeled temporal

slopes, and the red shading represents uncertainties in one standard deviation. The blue line is the mean of modeled

productivities, and the blue shading represents uncertainties in one standard deviation. In (c), the grey lines are the observed

temporal slopes, and the black line shows the observed spatial slope. Note that we simply converted observed ANPP from dry

mass (g DM m$^{-2}$ yr$^{-1}$) to carbon mass (g C m$^{-2}$ yr$^{-1}$) with a factor of 0.5.





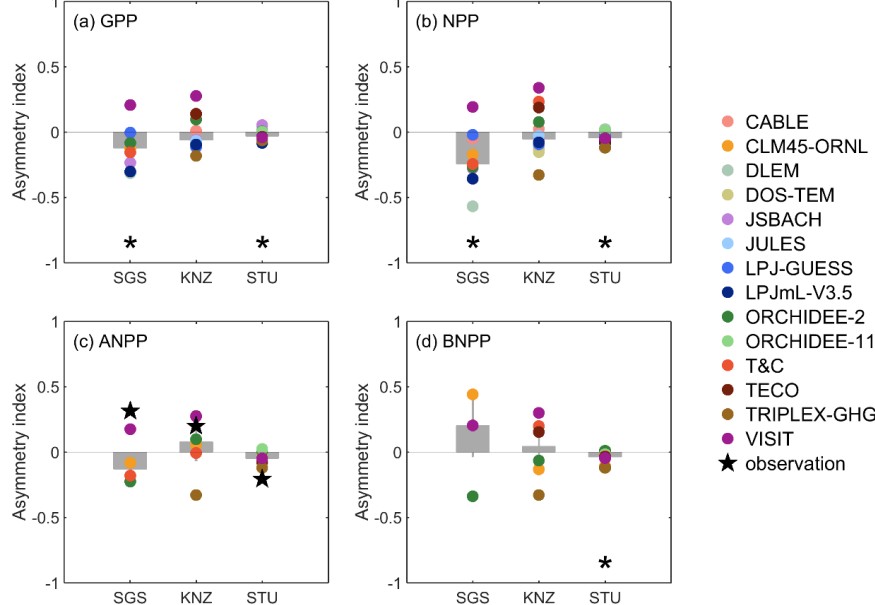

**Figure 4** Asymmetry responses of inter-annual GPP (a), NPP (b), ANPP (c) and BNPP (d) to precipitation in ambient simulations at the three sites SGS, KNZ and STU. The asymmetry index was calculated as the difference between the relative productivity pulses and declines in wet years and dry years (see Eq. (2) - Eq. (4)). Grey bars show median values among models, and black pentagrams in (c) represent asymmetry indices from observations. A black asterisk at the bottom of a panel indicates a significant asymmetry response of the model ensemble at 0.1 significance level by a non-parametric statistical hypothesis test.





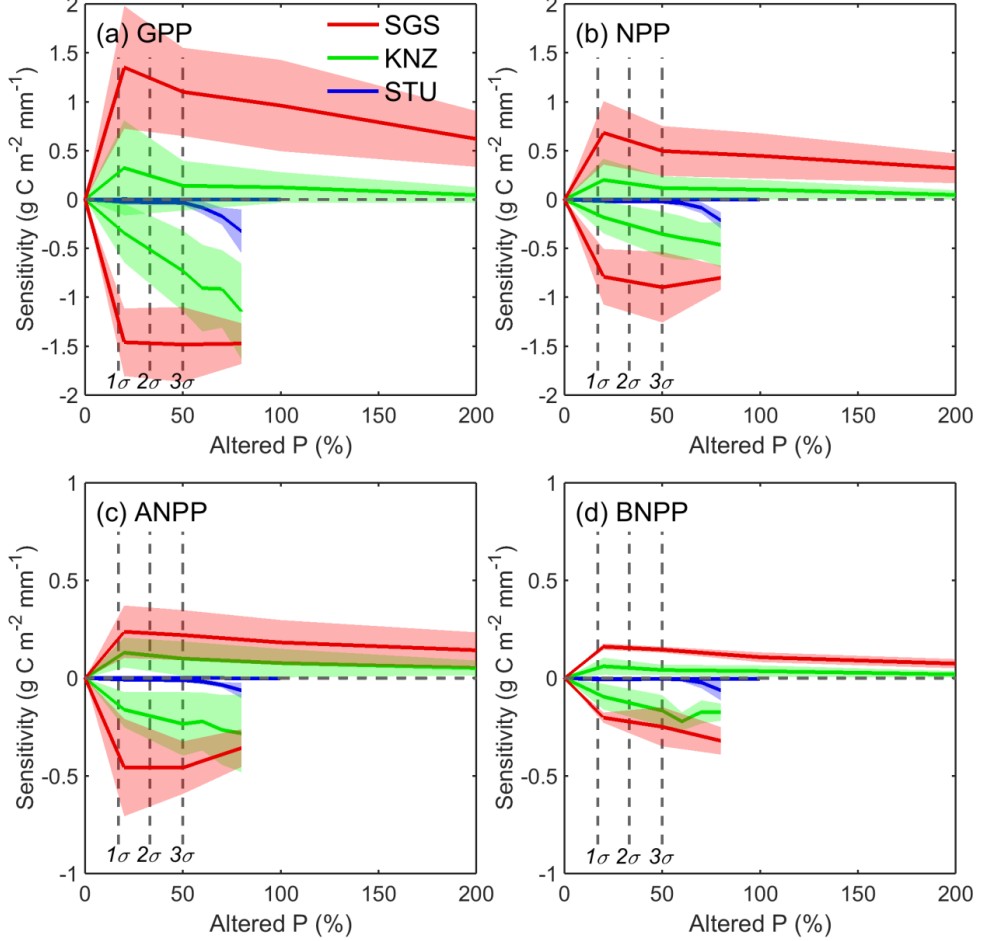

**Figure 5** Sensitivity of GPP (a), NPP (b), ANPP (c) and BNPP (d) for altered precipitation simulations at the three sites SGS, KNZ and STU. Curves show the median of models, and the shading represents uncertainty from median absolute deviation. Curves above the zero line represent responses under increasing precipitation conditions, and curves below the zero line show

5    responses under decreasing precipitation conditions relative to the control. Vertical dashed lines represent precipitation variations of one standard deviation (1σ, ~17% precipitation change), two standard deviations (2σ, ~33% precipitation change), and three standard deviations (3σ, ~50% precipitation change), which were derived from long-term annual precipitation at the three sites.



**Table 1** Key plant, soil, and climate characteristics of the three grassland sites. MAT, mean annual temperature; and MAP,

mean annual precipitation. MAT and MAP are based on the periods for the three sites with ANPP measurements.

|  | SGS | KNZ | STU |
|---|---|---|---|
| Latitude | 40°49′ N | 39°05′ N | 47°07′ N |
| Longitude | 104°46′ W | 96°35′ W | 11°19′ E |
| MAT(℃) | 8.6 | 13.0 | 6.2 |
| MAP(mm yr$^{-1}$) | 304 | 827 | 1429 |
| ANPP (g DM m$^{-2}$ yr$^{-1}$) | 91 | 387 | 525 |
| Measurement period | 1986-2009 | 1982-2012 | 2009-2013 |
| Grassland type | Shortgrass steppe | Mesic tallgrass prairie | Subalpine meadow |
| C3 species (%) | 30 | 15 | 100 |
| C4 species (%) | 70 | 85 | 0 |
| Soil type | Aridic Argiustoll | Typic Argiustoll | Dystric Cambisol |
| Sand (%) | 14 | 8 | 42 |
| Silt (%) | 58 | 60 | 31 |
| Clay (%) | 27 | 32 | 27 |





**Table 2** Summary of ecosystem models used in this study, including model name, nitrogen (N) cycle and relevant references. Also see Table S1-S14 for details of the simulated processes for grasslands in the ecosystem models, including N cycle, phosphorus (P) cycle, carbon (C) allocation scheme, carbohydrate reserves, leaf photosynthesis and stomatal conductance including treatment of water stress, scaling of photosynthesis from leaf to canopy, phenology, mortality, soil hydrology, surface
5  energy budget, root profile and dynamics, and grassland species.

| Model | Expanded Name | N cycle | References |
|---|---|---|---|
| CABLE | CSIRO Atmosphere Biosphere Land Exchange model | No | (Kowalczyk et al., 2006; Wang et al., 2011) |
| CLM45-ORNL | Version 4.5 of the Community Land Model | Yes | (Oleson et al., 2013) |
| DLEM | Dynamic Land Ecosystem Model | Yes | (Tian et al., 2011; Tian et al., 2015) |
| DOS-TEM | Dynamic Organic Soil structure in the Terrestrial Ecosystem Model | Yes | (Yi et al., 2010; McGuire et al., 1992) |
| JSBACH | Jena Scheme for Biosphere-Atmosphere Coupling in Hamburg | No | (Kaminski et al., 2013; Reick et al., 2013) |
| JULES | Joint UK Land Environment Simulator | No | (Best et al., 2011; Clark et al., 2011) |
| LPJ-GUESS | Lund-Potsdam-Jena General Ecosystem Simulator | Yes | (Smith et al., 2001; Smith et al., 2014a) |
| LPJmL-V3.5 | Lund-Potsdam-Jena managed Land | No | (Bondeau et al., 2007) |
| ORCHIDEE-2 | Organizing Carbon and Hydrology in Dynamic Ecosystems (2 soil layers) | No | (Krinner et al., 2005) |
| ORCHIDEE-11 | Organizing Carbon and Hydrology in Dynamic Ecosystems (11 soil layers) | No | (Krinner et al., 2005) |
| T&C | Tethys-Chloris | No | (Fatichi et al., 2012; Fatichi et al., 2016) |
| TECO | process-based Terrestrial Ecosystem model | No | (Weng and Luo, 2008) |
| TRIPLEX-GHG | An integrated process model of forest growth, carbon and greenhouse gases | Yes | (Peng et al., 2002; Zhu et al., 2014) |
| VISIT | Vegetation Integrative Simulator for Trace gases model | No | (Inatomi et al., 2010; Ito, 2010) |