# Peer review of "Asymmetric Responses of Primary Productivity to Altered Precipitation Simulated by Ecosystem Models across Three Longterm Grassland Sites"

_Biogeosciences, 2018_

## Referee Comment (RC1) · Anonymous Referee #1 · 19 Mar 2018

This manuscript presents a study of modeled and observed grassland NPP variability across years and three sites (Konza Prairie, Stubai Valley and the Central Plains Experimental Range). Fourteen terrestrial ecosystem models are used to simulate these three sites and the analysis focuses on modeled and observed responses of NPP to precipitation variability, and the asymmetry of these responses, i.e. different magnitude NPP responses for equivalent increases and decreases in precipitation. To assess model responses to precipitation variability in more detail, simulations that alter precipitation across a range of values at all three sites are also conducted.

In general the manuscript is well written and the simulations are extensive and well executed. The introduction is strong and well written with many references and fairly clear definition of goals. Given the organization of the abstract and the introduction, I find the presentation of results is not well structured. Also, the application of statistics could be improved. The interpretation of the results in the discussion is not well executed and the conclusions regarding mechanism are not well tied to a strong understanding of the mechanisms encoded in the models.

The abstract and introduction are structured around the asymmetry of responses and spatial versus temporal differences in responses. Yet the results are organized first with the simulations that alter precipitation (Figs 1 and 2); then the analysis of modeled and observed responses to spatial and temporal variation in precipitation (Figs 3 and 4); and then back to the model results of altered precipitation (Fig 5). I suggest leading with Figs 3 and 4, the comparison of model results and observations. Then follow with the altered precipitation results. The results from altered precipitation simulations should be used to interpret the model and observation comparison.

Uncertainty and statistics must be presented for the observations. Figs 3 and 4 need uncertainty bars on the observations. Why were stats done on the asymmetry of the model ensemble and not the observations. Statistical analysis of asymmetry in the observations must be done. In my view the stats on the model ensemble are unnecessary.

The discussion is poorly organized and in many cases only tangentially linked to the results presented in the study. The authors should rewrite the discussion, trying to avoid generality and link into their specific results.

There is an initial paragraph missing which summarizes the key result(s). The first paragraph of the discussion is mostly irrelevant, only the penultimate sentence relates to this particular study. The second paragraph of the discussion belongs in the results. The third paragraph is reasonable but should come later when trying to interpret the discrepancies between modeled and observed results. Section 4.2 is reasonable and

should lead the discussion, after the first paragraph suggested above.

p12 l28 – were these simulations done without using soil texture form the sites? That is a major oversight. It is clear that soil texture is a key driver of water availability in these models and could be a major reason for the discrepancy. If this were a single model study I would insist, but the logistics of a model intercomparison are such that I won't insist. Nevertheless, the authors should consider redoing the simulations with the actual soil textures. If they opt not to do this then it needs to be made very clear in the methods that site soil textures were not used in the study and the discussion should interpret the results in this context in more detail.

Section 4.3 is not really related to the study. None of the key conclusion in this section have been teased out from the analysis. Most of the recommendations are based on the literature cited in the section. What, in this study specifically, have you found the models are lacking and how can that be addressed? The authors need to do a much better job of identifying the short-falls in the models compared with the observations and providing a logical understanding of the causes of these shortfalls. Do some models perform better than others? If so, why?

The conclusions are poor. The first sentence stating novelty is unnecessary. The first half of the second sentence is not what I take from the results. Fig 4 shows the models do a bad job of capturing asymmetry at SGS but not at KNZ or STU. The second half of the second sentence is primarily speculation. The third sentence is a throwaway and is unnecessary. The fourth sentence seems to suggest that the collaboration between site investigators and modelers in this study was not very strong. The fifth sentence is about extremes, which this study only tangentially addresses, responses to "normal" variability are the focus of the study.

Minor comments:

p2 l3 – Unclear. Do you mean changes in variability or just variability? P2 l3-5 – Suggest moving this sentence to after the following one. P2 l5, l15, l20 etc – What

do you mean normally variable? P2 l8 – you switch between using asymmetrical and non-symmetrical, pick one and stick with it p2 l13-15 – awkward sentence, rephrase p2 l17 – what do you mean consistently here, across what? Sites? P2 l19-21 – Be more precise in what you mean here, what do you mean "extent of negative drought effects" and "impacts of increased precipitation" ? By "extent" and "impacts" do you mean different things? P3 l10 and many other places – "P-ANPP" sensitivities, you are analyzing ANPP in response to precipitation, it is conventional therefore to put ANPP first. Change to "ANPP-P". p3 l17-20 – this is a reasonable argument but depends on time-scale p4 l7 – quantify these rainfall regimes by adding MAP p4 l13-15 – this sentence is unnecessary, delete it or move to the discussion p4 l15-16 – this sentence is unnecessary, delete it p8 l15 – must add uncertainty to the observations p12 l8-10 – this sentence comes from nowhere, delete it p12 l28 – this is unclear, do you mean use measured SWC as an input? P12 l28 – rephrase "This will help in figuring out"

Figure 1. Differences in x-axis scales should be noted in the caption.

Figure 2. As above. Observations should be added to the ANPP plots.

Figure 3. Ho do you calculate a "mean slope"? Need uncertainty for observations. Technically standard deviation is not a measure of uncertainty, it is a measure of variability. I think your shading represents model variability.

Figure 4. The grey boxes are unnecessary, just use a black line if you want to show the mean/median. Change "pulses" to "gains." Need obs uncertainty.

Figure 5. "1 sigma ~17 %" was this the same across sites? Again, do you mean uncertainty or variability?

Table 1. Add variability (standard deviation) to MAT, MAP, and ANPP. If soil texture was not used in the simulations make this clear in the caption.

---

## Referee Comment (RC2) · Anonymous Referee #2 · 29 Mar 2018

The manuscript "Asymmetric Responses of Primary Productivity to Altered Precipitation Simulated by Ecosystem Models across Three Long- term Grassland Sites" presents a smart and well-thought out study to evaluate the performance of a large range of ecosystem models in their abilities to represent grassland productivity under changing climatic conditions. This study provides much needed insights in how ecosystem models perform when compared to field observations and highlights research needs to make such models more useful for climate change studies.

The abstract and introduction section is very nicely written and tightly structured. Unfortunately, I found that the result and discussion sections didn't follow this nice and

logical structure.

Most of the results do not fully account/present uncertainty estimates. Some do, but it is often insufficiently explained what measure of uncertainty/variation is presented. This makes it at time difficult to appropriately evaluate the relevance of patterns found in the results.

I miss an explicit discussion of the potential discrepancies in spatial scale between observations and model simulations. These can be particularly relevant for often fine-scale heterogeneity in soil moisture dynamics. I also miss a discussion on the caveats of the specific approach that was used for precipitation manipulations (fixed percentage increase/decrease for each rainfall event). It is not clear that this is what is happening under climate change; and precipitation event size distribution has large impacts on soil moisture dynamics (e.g., Lauenroth, W.K. & Bradford, J.B. (2012) Ecohydrology of dry regions of the United States: water balance consequences of small precipitation events. Ecohydrology, 5, 46–53.

Specific comments:

Introduction

- page 3, lines 13-15: rephrase to make the assumption explicit that "adaptation of plant communities over long time scales" is adaptation to typical "water received from rainfall for growth" – and not just any amount of water

- page 3, lines 13-17: placing all citations at the end can be interpreted that all these citations only support point 2 and that there is no citation to support point 1

- page 3, lines 17-20: The argument why temporal relationships are more informative for climate change impacts studies than spatial ones is not clear to me. It seems that effects of climate change on ANPP have not only a temporal trend (as stated here), but also include changes in species (and their adaptations) due to migration/extinction when tracking climate – thus spatial patterns may also be relevant if chosen carefully

to reflect projected climate differences.

Materials and Methods

- page 4, lines 19-26: I am surprised by the selection of the three study sites: two are located in the USA and represent naturally occurring grassland ecosystems where fire is an integral part whereas STU is located in Europe and is a man-made habitat that otherwise would be forested. These stark differences should at least be mentioned and caveats discussed.

- page 5, lines 24-25: Please provide some details on how the gap-filling was conducted and how much of the data were filled in – at least for precipitation. Various approaches can lead to considerable differences in precipitation values, e.g., seasonal biases in missing data.

- page 6, lines 20-25: * Why do you calculate the "median value of productivities in wet years with annual precipitation higher than the 90th percentile level" and don't simply take f(p95) = productivity value with annual precipitation at the 95th percentile? Aren't they the same? And equivalently for med(f(p10)) = f(p5)?

* It seems that AI simplifies to

** AI = (med(f(p90)) – mean(f)) / mean(f) – (mean(f) - med(f(p10)) / mean(f) # after inserting Rp and Rd and which simplifies to

** AI = (med(f(p90)) + med(f(p10))) / mean(f)

** AI = (f(p95) + f(p5)) / mean(f) # after inserting previous bullet point

* I don't understand why Rp and Rd are defined differently from each other and thus, AI is the sum instead of the difference between the 5%- and the 95%- quantiles. In most cases of somewhat symmetric distributions, f(p95) > mean(f) and mean(f) > f(p5) and thus AI > 0.

* Results presented for instance in Fig. 4 where AI < 0 and AI > 0 suggest that AI is

calculated correctly, as I suggest here, but that the equation is incorrectly written.

* What is meant with "f is the inter-annual productivity" (line 22)? Isn't f simply equal to "annual productivity"?

Results

- The structure of the result subsections is unexpected. The research questions and methods are tightly structured around the estimation of parameters a and b of Eq. 1, of the asymmetry index AI, and of the sensitivity index S. The result section does not follow this layout. For instance, the first subsection 3.1 could be presented in terms of estimation (and uncertainty) of parameter a. Then, the subsection 3.2 contains really the results (with lacking uncertainty estimates) for parameter b – plus in its current form some results on CUE and ANPP/NPP which have not been motived/introduced so far (which is confusing). The topic of subsection 3.3 spatial/temporal relationships presents the results from the second objective (as listed in the last paragraph of the introduction section); however, the method section does not explain how the observations and simulated values were aggregated and compared to address this question.

- page 7, lines 23-24: I see little support in Fig. 1 for the claim of a "steeper curvature at STU despite saturation above ambient precipitation indicates a steeper decline of productivity for precipitation set below ambient for this site compared to KNZ and SGS (Fig. 1)" – the precipitation treatment at STU did not (or at most barely) cover the curved part of the fitted lines. In most cases, a horizontal line appears to have fitted the data better. The estimates of b remain imprecise for STU, but this uncertainty is unfortunately not quantified.

- page 8, line 11: How was the ensemble model result calculated? Is this the arithmetic mean, median, etc.?

- page 8, line 23: "median value of -0.12±0.11" – what does the error component "±0.11" represent? Is this the MAD?

- page 8, line 23: Why "proportionally" larger? I don't understand what does could mean, particularly, because Rd and Rp are both calculated relative to mean(f).

- page 8, line 26: Why are the observed AI values presented without uncertainty estimate?

- Figure 4: The dots are too large relative to the figure; they are overlapping each other so much that it is really hard to see what is going on. For instance, the reported 0.1-"significance" with an unnamed test for STU seems dubious as the visible few dots huddle around 0.

- Figures S1 and S2: There are no error estimates for parameters a and b. At least add appropriate error bars to Figs S1-S2. I don't understand why a and b are presented against each other in a scatter plot. In my understanding, there is no expectation of a relationship between a and b. This is confusing.

- Figures S3 to S6: There is too much on these panels. It is no longer possible to identify responses of individual models.

- Figure S4: Is it correct that the "P" responses represents Rp of Eq. 3 and that "D" represents Rd of Eq. 4. Make this clear and use consistent terminology throughout the manuscript.

- Figures S5 and S6: How to the absolute SWC values compare between observed and simulated?

- Figures S7 to S9: Error estimates are missing and would be crucial to compare between CN- and C-only models.

Discussion

- page 9, lines 21-29: The first paragraph of the discussion reads like an introduction paragraph that identifies the knowledge gaps.

- page 11, lines 25-27: Not clear what is meant here with "arid and semi-arid grasses

[. . .] show relatively strong resistance". Does this refer to varying abilities of grass species to extract soil moisture held at increasingly higher tensions? If this were fixed values in models across sites, then the simulations models may produce too high sensitivities at the drier sites, particularly SGS.

- page 12, lines 1-2: I am confused here: the text continues to discuss "asymmetric responses" and yet refers to Fig. 5 which presents results for the sensitivity index calculated as relative difference among different model runs. So, if this text does refer to result for S, then I don't understand the statement "responses for normal precipitation variability" either because S isn't calculated from "normal precipitation variability" (as are Rd and Rp), but from manipulated precipitation inputs. Thus, S seems to rather represent sensitivity to deviation from 'normal' precipitation.

Data availability - Why are (a relevant subset of) data not deposited in a repository such as Dryad or figshare?

---

## Author Comment (AC1) · 3 May 2018

**Responses to the Comments of the Anonymous Referee #1**

**Please note that our responses are shown in black while the comments of the reviewers are in blue.**

This manuscript presents a study of modeled and observed grassland NPP variability across years and three sites (Konza Prairie, Stubai Valley and the Central Plains Experimental Range). Fourteen terrestrial ecosystem models are used to simulate these three sites and the analysis focuses on modeled and observed responses of NPP to precipitation variability, and the asymmetry of these responses, i.e. different magnitude NPP responses for equivalent increases and decreases in precipitation. To assess model responses to precipitation variability in more detail, simulations that alter precipitation across a range of values at all three sites are also conducted.

In general the manuscript is well written and the simulations are extensive and well executed. The introduction is strong and well written with many references and fairly clear definition of goals. Given the organization of the abstract and the introduction, I find the presentation of results is not well structured. Also, the application of statistics could be improved. The interpretation of the results in the discussion is not well executed and the conclusions regarding mechanism are not well tied to a strong understanding of the mechanisms encoded in the models.

**Responses:** We greatly appreciate the reviewer's pertinent feedback and valuable comments on our study. We thank you for your time and effort in helping us to improve this paper. In our revised version, we conducted a lot of work in responses to all these profound suggestions. In particular, we have (1) reorganized the main results in the abstract part; (2) reorganized the four specific objectives in the introduction part; (3) clarified the metrics of the response of productivity to precipitation changes following the four specific objectives in method part; (4) reorganized and strengthened the result part following the four specific objectives; (5) reorganized and strengthened the discussion part following the four specific objectives; and (6) rewrote the conclusion part to be more focused on our main results. We now consistently present first the comparison between spatial slopes and temporal slopes; which is followed by the asymmetric responses of productivities to precipitation under normal and extreme conditions using two indices (asymmetry index from inter-annual productivity and precipitation, and sensitivity of productivity to altered rainfall conditions). The curvilinear responses of productivities to altered precipitation across the three sites by each model are presented last. The first three specific objectives follow the structure used in Knapp et al. (2017), who have established a conceptual model for the precipitation-productivity relationships.

In addition, we have redrawn the figures to make them clearer, we have added uncertainty estimates for observation-based

asymmetry index, and we have also added uncertainty estimates by models. Detailed responses are as follows under each of your comments.

Knapp, A. K., Ciais, P., and Smith, M. D.: Reconciling inconsistencies in precipitation-productivity relationships: implications for climate change, New Phytologist, 214, 41-47, doi:10.1111/nph.14381, 2017.

5  The abstract and introduction are structured around the asymmetry of responses and spatial versus temporal differences in responses. Yet the results are organized first with the simulations that alter precipitation (Figs 1 and 2); then the analysis of modeled and observed responses to spatial and temporal variation in precipitation (Figs 3 and 4); and then back to the model results of altered precipitation (Fig 5). I suggest leading with Figs 3 and 4, the comparison of model results and observations. Then follow with the altered precipitation results. The results from altered precipitation simulations should be used to interpret

10  the model and observation comparison.

**Responses:** We agree. In our revised version, we have reorganized and strengthened the results following the same logical structure as the introductory section. We now consistently present first the comparison between spatial slopes and temporal slopes; which is followed by the asymmetric responses of productivities to precipitation under normal and extreme conditions using two indices (asymmetry index from inter-annual productivity and precipitation, and sensitivity of productivity to altered

15  rainfall conditions). The curvilinear responses of productivities to altered precipitation across the three sites by each model are presented last. The results have been improved in our revised version (page 8|line 8 to page 10|line 16).

Uncertainty and statistics must be presented for the observations. Figs 3 and 4 need uncertainty bars on the observations. Why were stats done on the asymmetry of the model ensemble and not the observations. Statistical analysis of asymmetry in the observations must be done. In my view the stats on the model ensemble are unnecessary.

20  **Responses:** We agree. In our revised version, we have added the uncertainty estimates of observation-based asymmetry index. The method has been clarified in our method part (page 6|line 24 to page 7|line 18).

In addition, we have added the uncertainty estimates of observation-based temporal slopes using a bootstrap sampling method with 1000 replicates of the ANPP and precipitation time series. We can obtain 1000 temporal slopes for each replicate and present the observed uncertainty ranges using interquartile spread of the temporal slopes between individual replicates ($10^{th}$

25  and $90^{th}$ percentiles).

The discussion is poorly organized and in many cases only tangentially linked to the results presented in the study. The authors should rewrite the discussion, trying to avoid generality and link into their specific results.

There is an initial paragraph missing which summarizes the key result(s). The first paragraph of the discussion is mostly irrelevant, only the penultimate sentence relates to this particular study. The second paragraph of the discussion belongs in the

results. The third paragraph is reasonable but should come later when trying to interpret the discrepancies between modeled and observed results. Section 4.2 is reasonable and should lead the discussion, after the first paragraph suggested above.

**Responses:** We find the criticism of the reviewer fair and we have rewrote the discussion section in the revised manuscript according to the comments raised. We firstly discussed the modeled and observed responses of productivity to altered precipitation, including spatial slopes and temporal slopes, and asymmetric responses of productivities to precipitation; we then discussed the curvilinear responses of productivities to altered precipitation by models; at last, we discussed the uncertainties of the two indices, knowledge gaps and suggestions of further work. We have removed the irrelevant discussion. The discussion have been improved in our revised version (page 10|line 17 to page 13|line 24).

p12 |28 – were these simulations done without using soil texture form the sites? That is a major oversight. It is clear that soil texture is a key driver of water availability in these models and could be a major reason for the discrepancy. If this were a single model study I would insist, but the logistics of a model intercomparison are such that I won't insist. Nevertheless, the authors should consider redoing the simulations with the actual soil textures. If they opt not to do this then it needs to be made very clear in the methods that site soil textures were not used in the study and the discussion should interpret the results in this context in more detail.

**Responses:** Our previous wording in the manuscript created a misunderstanding by the reviewer. Model simulations were carried out using soil texture properties measured at each site. All models used the same properties as reported in Table 1. We have clarified that in the revised manuscript. We totally agree that without soil textures from the sites the study would be weak, which is not the case.

Section 4.3 is not really related to the study. None of the key conclusion in this section have been teased out from the analysis. Most of the recommendations are based on the literature cited in the section. What, in this study specifically, have you found the models are lacking and how can that be addressed? The authors need to do a much better job of identifying the short-falls in the models compared with the observations and providing a logical understanding of the causes of these shortfalls. Do some models perform better than others? If so, why?

**Responses:** We agree with the referee and have thoroughly revised Section 4.3. We first discussed the uncertainties of the two indices (asymmetry index from inter-annual productivity and precipitation, and sensitivity of productivity to altered rainfall conditions) to study the asymmetric responses of productivities to precipitation under normal conditions. Then we indicate important knowledge gaps that should be considered in our following model-experiment interaction studies. This is the first study where a large group of modelers simulated the response of grassland primary productivity to precipitation using long-term observations for evaluating the asymmetry responses to altered precipitation. Hence, a lot of additional work is needed to deal with the very specific problems that became clear from our exercise. The Section 4.3 have been improved in our revised

version (page 13|line 1 to page 13|line 24).

The conclusions are poor. The first sentence stating novelty is unnecessary. The first half of the second sentence is not what I take from the results. Fig 4 shows the models do a bad job of capturing asymmetry at SGS but not at KNZ or STU. The second half of the second sentence is primarily speculation. The third sentence is a throwaway and is unnecessary. The fourth sentence seems to suggest that the collaboration between site investigators and modelers in this study was not very strong. The fifth sentence is about extremes, which this study only tangentially addresses, responses to "normal" variability are the focus of the study.

**Responses:** In our revised version, we have rewritten the conclusion part to tight it much closer to our main results. The conclusion has been improved in our revised version (page 13|line 27 to page 14|line 11).

Minor comments:

p2 |3 – Unclear. Do you mean changes in variability or just variability?

**Responses:** This sentence has been removed in our revised version.

P2 |3-5 – Suggest moving this sentence to after the following one.

**Responses:** This sentence stated the asymmetric responses of ANPP to precipitation under normal and extreme conditions observed from previous field measurements is a topic sentence in this paper. It also introduced the subject of this study here. Thus, we have left it unchanged.

P2 |5, |15, |20 etc – What do you mean normally variable?

**Responses:** In this study, normally variable means conditions of normal variation (not extreme) in precipitation.

P2 |8 – you switch between using asymmetrical and non-symmetrical, pick one and stick with it

**Responses:** This expression has been revised to asymmetric.

p2 |13-15 – awkward sentence, rephrase

**Responses:** This sentence has been revised to: "The spatial slopes derived from modeled primary productivity and precipitation across sites were steeper than the temporal slopes obtained from inter-annual variations, which was consistent with empirical data."

p2 |17 – what do you mean consistently here, across what? Sites?

**Responses:** "consistently" has been removed in our revised version.

 – Be more precise in what you mean here, what do you mean "extent of negative drought effects" and "impacts of increased precipitation" ? By "extent" and "impacts" do you mean different things?

**Responses:** Here, we meant the same thing. This sentence has been revised to: "Our results indicated that most models overestimate the negative drought effects and/or underestimate the positive effects of increased precipitation on primary productivity under normal climate conditions, highlighting the need for improving eco-hydrological processes in those models in the future."

P3 |10 and many other places – "P-ANPP" sensitivities, you are analyzing ANPP in response to precipitation, it is conventional therefore to put ANPP first. Change to "ANPP-P".

**Responses:** This expression has been revised to "ANPP-P".

p3 |17-20 – this is a reasonable argument but depends on time-scale

**Responses:** We agree, and we also expressed the idea here. To make it more clear, this sentence has been revised to: "For projecting the effect of climate change on grassland productivity in near to mid-term (coming decades), inter-annual relationships are arguably more informative than spatial relationships because spatial relationships reflect long-term adaptation of ecosystems, and because ANPP-P relationships from spatial gradients are confounded by the co-variation of gradients in other environmental variables (e.g. temperature and radiation) and soil properties (Estiarte et al., 2016; Knapp et al., 2017b)."

p4 |7 – quantify these rainfall regimes by adding MAP

**Responses:** This sentence has been revised to: "In this study, we aim to evaluate the responses of simulated productivity to altered precipitation from fourteen ecosystem models at three sites representing dry ($304 \pm 118$ mm yr$^{-1}$), mesic ($827 \pm 175$ mm yr$^{-1}$), and moist ($1429 \pm 198$ mm yr$^{-1}$) rainfall regimes."

p4 |13-15 – this sentence is unnecessary, delete it or move to the discussion

**Responses:** This sentence has been removed.

p4 |15-16 – this sentence is unnecessary, delete it

**Responses:** This sentence has been removed.

p8 |15 – must add uncertainty to the observations

**Responses:** Uncertainty for observation-based results has been added in our revised version.

p12 |8-10 – this sentence comes from nowhere, delete it

**Responses:** This sentence has been removed.

p12 |28 – this is unclear, do you mean use measured SWC as an input?

**Responses:** In this sentence, we mean that models should simulate SWC in the same soil layer as experiments in following studies, so that we could evaluate the modeled SWC compared to observations. This sentence has been revised to: "Models should report SWC at the same depth of experiments and experimental data should be made available for better comparisons in following studies. This can provide insights into the bias of modeled sensitivities to precipitation and check explicitly the sensitivity of vegetation productivity to change in SWC"

P12 |28 – rephrase "This will help in figuring out"

**Responses:** This expression has been removed.

Figure 1. Differences in x-axis scales should be noted in the caption.

**Responses:** This has been added in our revised version (page 26).

Figure 2. As above. Observations should be added to the ANPP plots.

**Responses:** This has been added in our revised version (page 27). Observations are shown in our first figure (temporal slope vs. spatial slope) in our revised version, and the curvilinear responses of productivities to altered precipitation by models have been moved to the last part.

Figure 3. Ho do you calculate a "mean slope"? Need uncertainty for observations. Technically standard deviation is not a measure of uncertainty, it is a measure of variability. I think your shading represents model variability.

**Responses:** Here, for each site, we firstly calculated the temporal slopes for each model under ambient simulation relating inter-annual variability in precipitation to inter-annual variability in the productivities using linear regression analysis. Then, we could calculate a mean temporal slope. In addition, we have presented the model uncertainty ranges using interquartile spread of the temporal slopes between individual simulations ($10^{th}$ and $90^{th}$ percentiles). The uncertainty estimates for the observation have been added in our revised version (page 23).

Figure 4. The grey boxes are unnecessary, just use a black line if you want to show the mean/median. Change "pulses" to "gains." Need obs uncertainty.

**Responses:** The grey boxes are removed in our revised figure. Here, we chose similar expression as that in Knapp and Smith (2001) (relative ANPP pulse and relative ANPP decline). Thus, we have left it unchanged. The uncertainty estimates for the observation have been added in our revised version (page 24).

Knapp, A. K., and Smith, M. D.: Variation among Biomes in Temporal Dynamics of Aboveground Primary Production, Science, 291, 481-484, doi:10.1126/science.291.5503.481, 2001.

Figure 5. "1 sigma ~ 17 %" was this the same across sites? Again, do you mean uncertainty or variability?

**Responses:** In our revised figure, we have presented the variability of the three sites individually (1 sigma, 2 sigma and 3 sigma). In addition, we have presented the model uncertainty ranges using interquartile spread of the sensitivities between individual simulations (10th and 90th percentiles) (page 25).

Table 1. Add variability (standard deviation) to MAT, MAP, and ANPP. If soil texture was not used in the simulations make this clear in the caption.

**Responses:** The variability has been added in our revised version. In ecosystem model simulations, models have used the soil textures from the three sites as described in Table 1 (page 28).

---

## Author Comment (AC2) · 3 May 2018

5 **Responses to the Comments of the Anonymous Referee #2**

**Please note that our responses are shown in black while the comments of the reviewers are in blue.**

The manuscript "Asymmetric Responses of Primary Productivity to Altered Precipitation Simulated by Ecosystem Models across Three Long- term Grassland Sites" presents a smart and well-thought out study to evaluate the performance of a large range of ecosystem models in their abilities to represent grassland productivity under changing climatic conditions. This study

10 provides much needed insights in how ecosystem models perform when compared to field observations and highlights research needs to make such models more useful for climate change studies.

The abstract and introduction section is very nicely written and tightly structured. Unfortunately, I found that the result and discussion sections didn't follow this nice and logical structure.

**Responses:** We greatly appreciate the reviewer's pertinent feedback and valuable comments on our study. We thank you for

15 your time and effort in helping us to improve this paper. In our revised version, we conducted a lot of work in responses to all these profound suggestions. In particular, we have (1) reorganized the main results in the abstract part; (2) reorganized the four specific objectives in the introduction part; (3) clarified the metrics of the response of productivity to precipitation changes following the four specific objectives in method part; (4) reorganized and strengthened the result part following the four specific objectives; (5) reorganized and strengthened the discussion part following the four specific objectives; and (6) rewrote

20 the conclusion part to be more focused on our main results. We now consistently present first the comparison between spatial slopes and temporal slopes; which is followed by the asymmetric responses of productivities to precipitation under normal and extreme conditions using two indices (asymmetry index from inter-annual productivity and precipitation, and sensitivity of productivity to altered rainfall conditions). The curvilinear responses of productivities to altered precipitation across the three sites by each model are presented last. The first three specific objectives follow the structure used in Knapp et al. (2017), who

25 have established a conceptual model for the precipitation-productivity relationships.

In addition, we have redrawn the figures to make them clearer, we have added uncertainty estimates for observation-based asymmetry index, and we have also added uncertainty estimates by models. Detailed responses are as follows under each of your comments.

Knapp, A. K., Ciais, P., and Smith, M. D.: Reconciling inconsistencies in precipitation-productivity relationships: implications

for climate change, New Phytologist, 214, 41-47, doi:10.1111/nph.14381, 2017.

Most of the results do not fully account/present uncertainty estimates. Some do, but it is often insufficiently explained what measure of uncertainty/variation is presented. This makes it at time difficult to appropriately evaluate the relevance of patterns found in the results.

**Responses:** In our revised results and figures, we have added uncertainty estimates by models, which presenting the model uncertainty ranges using interquartile spread of the sensitivities between individual simulations (10th and 90th percentiles).

In addition, we have added the uncertainty estimates of observation-based asymmetry index. The method has been clarified in our method part (page 6|line 24 to page 7|line 18). We have also added the uncertainty estimates of observation-based temporal slopes using a bootstrap sampling method with 1000 replicates of the ANPP and precipitation time series. We can obtain 1000 temporal slopes for each replicate and present the observed uncertainty ranges using interquartile spread of the temporal slopes between individual replicates (10th and 90th percentiles).

I miss an explicit discussion of the potential discrepancies in spatial scale between observations and model simulations. These can be particularly relevant for often fine-scale heterogeneity in soil moisture dynamics. I also miss a discussion on the caveats of the specific approach that was used for precipitation manipulations (fixed percentage increase/decrease for each rainfall event). It is not clear that this is what is happening under climate change; and precipitation event size distribution has large impacts on soil moisture dynamics (e.g., Lauenroth, W.K. & Bradford, J.B. (2012) Ecohydrology of dry regions of the United States: water balance consequences of small precipitation events. Ecohydrology, 5, 46–53.

**Responses:** We thank the referee for this insightful remark. In our revised version, we have added discussion on these issues in the second paragraph of section 4.3 related to the uncertainties, knowledge gaps and suggestions of future work. The second paragraph of section 4.3 have been revised as follows:

Although the carbon-water interactions in current models have been improved during the last decades, there still exist large gaps for accurately diagnosing the errors in the representation of key processes and parameterizations. Suggestions that should be considered in future studies aimed at model-data interaction include: (1) Simulation of SWC in the soil layer(s) for which experimental data are available. This can provide insights into the bias of modeled sensitivities to precipitation and check explicitly the sensitivity of vegetation productivity to change in SWC; (2) more experiments are needed that assess also BNPP in order to evaluate the corresponding processes in models (Luo et al., 2017; Wilcox et al., 2017); (3) there still exist large gaps between changes of precipitation occurrence and intensity in reality and how we simulated them in the current work, i.e., the altered rainfall forcing datasets were constructed by decreasing/increasing the amount of precipitation in each precipitation event by a fixed percentage during the time-span of productivity observations at each site and not by modifying precipitation

structure or reproducing the real treatment. Further studies need to consider better different scenarios of precipitation occurrence and intensity under climate change (Lauenroth and Bradford, 2012), which will likely help to better understand the responses of productivities to altered precipitation in the next decades. In addition, modelers will need to simulate the control experiments corresponding to the real local precipitation manipulations applied by field scientists, e.g., considering the observed time series of modified precipitation and vegetation composition, root profiles, nutrient cycling, phenology and carbon allocation as close as possible to local conditions. This should be a priority for future model-experiment interaction studies.

Specific comments:

Introduction

- page 3, lines 13-15: rephrase to make the assumption explicit that "adaptation of plant communities over long time scales" is adaptation to typical "water received from rainfall for growth" – and not just any amount of water

**Responses:** The expression has been revised to: "a 'vegetation constraint' reflecting the adaptation of plant communities over long time scales in such a way that grasslands make the best use of the typical water received from rainfall for growth."

- page 3, lines 13-17: placing all citations at the end can be interpreted that all these citations only support point 2 and that there is no citation to support point 1

**Responses:** The citations have been revised.

- page 3, lines 17-20: The argument why temporal relationships are more informative for climate change impacts studies than spatial ones is not clear to me. It seems that effects of climate change on ANPP have not only a temporal trend (as stated here), but also include changes in species (and their adaptations) due to migration/extinction when tracking climate – thus spatial patterns may also be relevant if chosen carefully to reflect projected climate differences.

**Responses:**

Here, we preferred the temporal models than spatial models for projecting the effect of climate change on grassland productivity because substantial changes in plant communities (turnover of dominant life-forms) and corresponding alterations in soil biogeochemistry only occur over long time scales (decades to centuries). However, even over long time scales, the future climate and interactions with other global change drivers are expected to lead to communities that do not match current spatially observed climate-vegetation patterns. Thus, at least for near to mid-term forecasts of climate change effects on productivities, temporal models are more representative than spatial models (Knapp et al., 2017). In our revised version, the sentence has been revised to: "For projecting the effect of climate change on grassland productivity in near to mid-term (coming

decades), inter-annual relationships are arguably more informative than spatial relationships because spatial relationships reflect long-term adaptation of ecosystems, and because ANPP-P relationships from spatial gradients are confounded by the co-variation of gradients in other environmental variables (e.g. temperature and radiation) and soil properties (Estiarte et al., 2016; Knapp et al., 2017)."

5   Knapp, A. K., Ciais, P., and Smith, M. D.: Reconciling inconsistencies in precipitation-productivity relationships: implications for climate change, New Phytologist, 214, 41-47, doi:10.1111/nph.14381, 2017.

Materials and Methods

- page 4, lines 19-26: I am surprised by the selection of the three study sites: two are located in the USA and represent naturally occurring grassland ecosystems where fire is an integral part whereas STU is located in Europe and is a man-made habitat that

10  otherwise would be forested. These stark differences should at least be mentioned and caveats discussed.

**Responses:** These three grasslands were selected because they lie along a mean annual precipitation (MAP) gradient, and have detailed meteorological data to force the models. While two are "natural" grasslands (KNZ and SGS) and one (STU) is not, global land surface models do not typically differentiate regarding the origin of ecosystem types and heavily managed grasslands and pastures represent a significant fraction of mesic grasslands globally. Semi-natural subalpine grasslands in the

15  Alps were created several centuries ago, are very lightly managed and should be in equilibrium concerning soil physical conditions. It should be noted though that the grassland at STU is cut once a year and lightly fertilized every 2-4 years and in consequence differs in plant composition and soil fungi: bacteria ratio, which leads to different drought responses compared to abandoned grassland (Ingrisch et al., 2017; Karlowsky et al., 2018). Further, it is worth noting that the mesic grassland in the USA would also be forested if human-initiated prescribed fires were to be removed from the system (Briggs et al. 2005).

20  Thus, these grassland sites lie along a continuum of dry natural grassland, mesic natural grassland maintained by human management, and anthropogenic moist grassland maintained by human management. We now use these descriptions when the sites are initially described.

Ingrisch, J., Karlowsky, S., Anadon-Rosell, A., Hasibeder, R., König, A., Augusti, A., Gleixner, G., and Bahn, M.: Land Use Alters the Drought Responses of Productivity and $CO_2$ Fluxes in Mountain Grassland, Ecosystems, doi:10.1007/s10021-017-

25  0178-0, 2017.

Karlowsky, S., Augusti, A., Ingrisch, J., Hasibeder, R., Lange, M., Lavorel, S., Bahn, M., Gleixner, G., and Wurzburger, N.: Land use in mountain grasslands alters drought response and recovery of carbon allocation and plant-microbial interactions, Journal of Ecology, 106, 1230-1243, doi:10.1111/1365-2745.12910, 2018.

Briggs, J. M., Knapp, A. K., Blair, J. M., Heisler, J. L., Hoch, G. A., Lett, M. S., and McCARRON, J. K.: An ecosystem in

transition: causes and consequences of the conversion of mesic grassland to shrubland, BioScience, 55, 243-254, doi:10.1641/0006-3568(2005)055[0243:AEITCA]2.0.CO;2, 2005.

- page 5, lines 24-25: Please provide some details on how the gap-filling was conducted and how much of the data were filled in – at least for precipitation. Various approaches can lead to considerable differences in precipitation values, e.g., seasonal biases in missing data.

**Responses:** Historical reconstructions of meteorological variables from gridded CRUNCEP data at 1/2 hourly time step, a reanalysis product that does not have gaps (Wei et al., 2014), were combined and bias-corrected with site observations to provide bias corrected historical forcing time series from 1901 to 2013 (CRUNCEP-BC). For example, the CRUNCEP precipitation data will be regressed against the SGS precipitation observations during 1986-2009 as $y=a*x+b$, where y is the SGS precipitation observations during 1986-2009; x is the CRUNCEP precipitation data at SGS during 1986-2009. Then, the long term forcing data in SGS can be produced by correcting CRUNCEP data (called CRUNCEP-BC, BC is for bias corrected) during 1901-2013 using the equation and fitted parameters of $a$ and $b$.

- page 6, lines 20-25: * Why do you calculate the "median value of productivities in wet years with annual precipitation higher than the 90th percentile level" and don't simply take f(p95) = productivity value with annual precipitation at the 95th percentile? Aren't they the same? And equivalently for med(f(p10)) = f(p5)?

* It seems that AI simplifies to

** AI = (med(f(p90)) – mean(f)) / mean(f) – (mean(f) - med(f(p10)) / mean(f) # after inserting Rp and Rd and which simplifies to

** AI = (med(f(p90)) + med(f(p10))) / mean(f)

** AI = (f(p95) + f(p5)) / mean(f) # after inserting previous bullet point

* I don't understand why Rp and Rd are defined differently from each other and thus, AI is the sum instead of the difference between the 5%- and the 95%- quantiles. In most cases of somewhat symmetric distributions, f(p95) > mean(f) and mean(f) > f(p5) and thus AI > 0.

* Results presented for instance in Fig. 4 where AI < 0 and AI > 0 suggest that AI is calculated correctly, as I suggest here, but that the equation is incorrectly written.

* What is meant with "f is the inter-annual productivity" (line 22)? Isn't f simply equal to "annual productivity"?

**Responses:**

In this method, for example, we chose the $med(f_{p90})$, the median value of productivities in wet years with annual precipitation higher than the $90^{th}$ percentile level, rather than directly using the value of productivity in wet year at the $95^{th}$ percentile level because the latter method may produce an extreme value of productivity which is less representative of the relative productivity pulse in wet years for the particular site. In other words, we opted for this approach to avoid artefacts.

5     In order to characterize the asymmetry of productivity to precipitation, we define the asymmetry index (AI) from inter-annual productivity and precipitation data as follows:

$$AI = R_p - R_d \tag{1}$$

where $R_p$ is the relative productivity pulse in wet years, and $R_d$ is the relative productivity decline in dry years defined by:

$$R_p = (med(f_{p90}) - \bar{f})/\bar{f} \tag{2}$$

10     $$R_d = (\bar{f} - med(f_{p10}))/\bar{f} \tag{3}$$

Thus, the *AI* could be rewritten as follows:

$$AI = \frac{med(f_{p90}) - \bar{f}}{\bar{f}} - \frac{\bar{f} - med(f_{p10})}{\bar{f}} = \frac{med(f_{p90}) + med(f_{p10}) - 2*\bar{f}}{\bar{f}} \tag{4}$$

where $f$ is the inter-annual productivity, being a function of environmental factors from models or observation; $\bar{f}$ is mean annual productivity in the period of measurements (Table 1); $med(f_{p90})$ is the median value of productivities in wet years

15     with annual precipitation higher than the $90^{th}$ percentile level; $med(f_{p10})$ is median value of productivities in all the dry years when annual precipitation is lower than the $10^{th}$ percentile level.

In general, $R_p > 0$ indicates that the median value of productivities in wet years is higher than the mean annual productivity in the period of measurements; and $R_d > 0$ indicates that the median value of productivities in dry years is smaller than the mean annual productivity in the period of measurements. Therefore, if *AI* > 0, i.e., a positive asymmetry, means that there is a

20     greater increase of productivity in wet years than decline in dry years; if *AI* < 0, i.e., a negative asymmetry, means that there is a greater decline of productivity in dry years than increase in wet years.

In our revised version, we have expanded the description of the asymmetry index to make this clear for readers.

Results

- The structure of the result subsections is unexpected. The research questions and methods are tightly structured around the

25     estimation of parameters a and b of Eq. 1, of the asymmetry index AI, and of the sensitivity index S. The result section does not follow this layout. For instance, the first subsection 3.1 could be presented in terms of estimation (and uncertainty) of parameter a. Then, the subsection 3.2 contains really the results (with lacking uncertainty estimates) for parameter b – plus in

its current form some results on CUE and ANPP/NPP which have not been motived/introduced so far (which is confusing). The topic of subsection 3.3 spatial/temporal relationships presents the results from the second objective (as listed in the last paragraph of the introduction section); however, the method section does not explain how the observations and simulated values were aggregated and compared to address this question.

**Responses:** In our revised version, we have reorganized and strengthened the results following the same logical structure as the introductory section. We now consistently present first the comparison between spatial slopes and temporal slopes; which is followed by the asymmetric responses of productivities to precipitation under normal and extreme conditions using two indices (asymmetry index from inter-annual productivity and precipitation, and sensitivity of productivity to altered rainfall conditions). The curvilinear responses of productivities to altered precipitation across the three sites by each model are presented last. The results have been improved in our revised version (page 8|line 8 to page 10|line 16).

- page 7, lines 23-24: I see little support in Fig. 1 for the claim of a "steeper curvature at STU despite saturation above ambient precipitation indicates a steeper decline of productivity for precipitation set below ambient for this site compared to KNZ and SGS (Fig. 1)" – the precipitation treatment at STU did not (or at most barely) cover the curved part of the fitted lines. In most cases, a horizontal line appears to have fitted the data better. The estimates of b remain imprecise for STU, but this uncertainty is unfortunately not quantified.

**Responses:** In our revised version, we have redrawn the figures to make it clear for readers (page 26). The fitted lines only covered the simulations under altered precipitation conditions. In addition, the expression "steeper curvature at STU despite saturation above ambient precipitation indicates a steeper decline of productivity for precipitation set below ambient for this site compared to KNZ and SGS" has been removed.

- page 8, line 11: How was the ensemble model result calculated? Is this the arithmetic mean, median, etc.?

- page 8, line 23: "median value of -0.12±0.11" – what does the error component "±0.11" represent? Is this the MAD?

**Responses:** The ensemble model result used the arithmetic mean value, and the model uncertainty ranges used interquartile spread of the asymmetry indices between individual simulations (10th and 90th percentiles) in our revised version.

- page 8, line 23: Why "proportionally" larger? I don't understand what does could mean, particularly, because Rd and Rp are both calculated relative to mean(f).

**Responses:** This sentence has been revised to: "Hence, for SGS simulated declines of GPP and NPP in dry years were larger than the increases in wet years."

- page 8, line 26: Why are the observed AI values presented without uncertainty estimate?

**Responses:** We agree. In our revised version, we have added the uncertainty estimates of observation-based asymmetry index. The method has been clarified in our method part (page 6|line 24 to page 7|line 18).

- Figure 4: The dots are too large relative to the figure; they are overlapping each other so much that it is really hard to see what is going on. For instance, the reported 0.1-"significance" with an unnamed test for STU seems dubious as the visible few dots huddle around 0.

**Responses:** In our revised version, we have redrawn the figure to make it clear for readers (page 24). Here, a nonparametric test called the Wilcoxon signed rank was used for the significance test.

- Figures S1 and S2: There are no error estimates for parameters a and b. At least add appropriate error bars to Figs S1-S2. I don't understand why a and b are presented against each other in a scatter plot. In my understanding, there is no expectation of a relationship between a and b. This is confusing.

**Responses:** In our revised figures, we have added error estimates for parameters *a* and *b*. Here, we just intended to present the parameters *a* and *b* in a two-dimensional space rather than to make a relationship between the two parameters.

- Figures S3 to S6: There is too much on these panels. It is no longer possible to identify responses of individual models.

**Responses:** In our revised version, we have redrawn the figure to make it clear for readers.

- Figure S4: Is it correct that the "P" responses represents Rp of Eq. 3 and that "D" represents Rd of Eq. 4. Make this clear and use consistent terminology throughout the manuscript.

**Responses:** In our revised version, we changed the "P" and "D" to "$R_p$" and "$R_d$" in the figure to make the expression consistently throughout the manuscript.

- Figures S5 and S6: How to the absolute SWC values compare between observed and simulated?

**Responses:** The two figures are used to discuss the curvilinear responses of productivities to altered precipitation by models rather than to compare with observations. In this first model-experiment interaction study, we did not have any observed SWC data under differently altered precipitation conditions. For comparing the observed and modeled SWC, we recommended that models should report SWC at the same depth of experiments and experimental data should be made available for better comparisons in following studies. This can provide insights into the bias of modeled sensitivities to precipitation and check explicitly the sensitivity of vegetation productivity to change in SWC.

- Figures S7 to S9: Error estimates are missing and would be crucial to compare between CN- and C-only models.

**Responses:** In our revised figures, we have added uncertainty estimates for C-N and C-only models, which presenting the

model uncertainty ranges using interquartile spread of the sensitivities between individual simulations (10th and 90th percentiles).

- page 9, lines 21-29: The first paragraph of the discussion reads like an introduction paragraph that identifies the knowledge gaps.

**Responses:** This paragraph has been removed in our revised version.

- page 11, lines 25-27: Not clear what is meant here with "arid and semi-arid grasses […] show relatively strong resistance". Does this refer to varying abilities of grass species to extract soil moisture held at increasingly higher tensions? If this were fixed values in models across sites, then the simulations models may produce too high sensitivities at the drier sites, particularly SGS.

**Responses:** Grassland root depth affects ecosystem resilience to environmental stress such as drought, and arid and semi-arid grasses that have extensive lateral roots or possibly deep roots show relatively strong resistance. We agree, this also means that arid and semi-arid grasses may show greater abilities to extract soil moisture under conditions of increasing water stress. However, most ecosystem models currently consider only two types of grasslands, C3 and C4 (Table S14) with fixed root profiles along with prescribed soil layers (Table S13). This is potentially unrealistic for semi-arid grass roots and can lead to models underestimating the accessible water and the resistance to drought. We agree, the models may also produce too high sensitivities to drought at the drier sites.

- page 12, lines 1-2: I am confused here: the text continues to discuss "asymmetric responses" and yet refers to Fig. 5 which presents results for the sensitivity index calculated as relative difference among different model runs. So, if this text does refer to result for S, then I don't understand the statement "responses for normal precipitation variability" either because S isn't calculated from "normal precipitation variability" (as are Rd and Rp), but from manipulated precipitation inputs. Thus, S seems to rather represent sensitivity to deviation from 'normal' precipitation.

**Responses:** In this work, we characterized the asymmetric responses of productivities to precipitation under normal and extreme conditions using two indices (asymmetry index from inter-annual productivity and precipitation, and sensitivity of productivity to altered rainfall conditions). The asymmetry index could present the asymmetric responses under normal conditions, and the sensitivity of productivity to altered rainfall conditions could suggest the asymmetric responses under both normal and extreme conditions (Knapp et al., 2017). In our revised version, the sentence has been revised to "The sensitivity of productivity to increased and decreased precipitation for simulations where mean precipitation was normally altered generally suggested negative asymmetric responses at dry (SGS) and mesic (KNZ) sites (Fig. 3c)."

**Responses:** The amount of data is very large (about 25 GB), and we will try to use an online repository for sharing the modeled outputs.

outputs.

---

## Author Response (AR1)

**Responses to the Comments of the Associate Editor**

**Please note that our responses are shown in black while the comments are in blue.**

Comments to the Author:

Dear Donghai Wu,

Thank you for your thoughtful response to the reviewer comments and the improvements you have suggested. I look forward to receiving the revised manuscript. I would also encourage you to follow the reviewer's suggestion to host the data used in the study in a publicly accessible repository. 25GB is a lot, and too much for sites such as GitHub, but surely the key variables at an appropriate temporal resolution do not represent such a large file.

Best wishes,

Trevor

**Responses:**

Dear Dr. Trevor Keenan,

Thank you very much for your great efforts on our manuscript. We also greatly thank the two anonymous reviewers for providing pertinent feedback and valuable comments that greatly improved the quality of this paper. We carefully revised the manuscript according to all these profound comments, and hope that the revised version could satisfy you and the reviewers. In addition, we have upload the data to a publicly accessible repository. All the modeled outputs in the first model-experiment interaction study can be publicly obtained from https://pan.baidu.com/s/1CXAnStQMBD_4a0tLGiIpiQ.

Please contact us if any further information is requested. In advance, thank you for your consideration.

Best wishes to you!

Sincerely yours,

Donghai Wu (On behalf of all of the co-authors)

**List of all relevant changes made in the manuscript**

In particular, we have (1) reorganized the main results in the abstract part; (2) reorganized the four specific objectives in the introduction part; (3) clarified the metrics of the response of productivity to precipitation changes following the four specific objectives in method part; (4) reorganized and strengthened the result part following the four specific objectives; (5) reorganized and strengthened the discussion part following the four specific objectives; and (6) rewrote the conclusion part to be more focused on our main results. We now consistently present first the comparison between spatial slopes and temporal slopes; which is followed by the asymmetric responses of productivities to precipitation under normal and extreme conditions using two indices (asymmetry index from inter-annual productivity and precipitation, and sensitivity of productivity to altered rainfall conditions). The curvilinear responses of productivities to altered precipitation across the three sites by each model are presented last. The first three specific objectives follow the structure used in Knapp et al. (2017), who have established a conceptual model for the precipitation-productivity relationships.

In addition, we have redrawn the figures to make them clearer, we have added uncertainty estimates for observation-based asymmetry index, and we have also added uncertainty estimates by models. In addition, we have upload the data to a publicly accessible repository. All the modeled outputs in the first model-experiment interaction study can be publicly obtained from https://pan.baidu.com/s/1CXAnStQMBD_4a0tLGiIpiQ.

In following mask-up manuscript version, we changed the major revised contents as red color. Very detailed revises (words, sentences, reorganized and deleted contents, and figures in supplement) have been clearly reported in the reports of "Responses to the Comments of the Anonymous Referee #1" and "Responses to the Comments of the Anonymous Referee #2" following the reviewers' comments.

**Asymmetric Responses of Primary Productivity to Altered Precipitation Simulated by Ecosystem Models across Three Long-term Grassland Sites**

Donghai Wu[1], Philippe Ciais[2], Nicolas Viovy[2], Alan K. Knapp[3], Kevin Wilcox[4], Michael Bahn[5], Melinda D. Smith[3], Sara Vicca[6], Simone Fatichi[7], Jakob Zscheischler[8], Yue He[1], Xiangyi Li[1], Akihiko Ito[9], Almut Arneth[10], Anna Harper[11], Anna Ukkola[12], Athanasios Paschalis[13], Benjamin Poulter[14], Changhui Peng[15,16], Daniel Ricciuto[17], David Reinthaler[5], Guangsheng Chen[18], Hanqin Tian[18], Hélène Genet[19], Jiafu Mao[17], Johannes Ingrisch[5], Julia E.S.M. Nabel[20], Julia Pongratz[20], Lena R. Boysen[20], Markus Kautz[10], Michael Schmitt[5], Patrick Meir[21,22], Qiuan Zhu[16], Roland Hasibeder[5], Sebastian Sippel[23], Shree R.S. Dangal[18,24], Stephen Sitch[25], Xiaoying Shi[17], Yingping Wang[26], Yiqi Luo[4,27], Yongwen Liu[1], Shilong Piao[1]

[1] Sino-French Institute for Earth System Science, College of Urban and Environmental Sciences, Peking University, Beijing, 100871, China.

[2] Laboratoire des Sciences du Climat et de l'Environnement, CEA-CNRS-UVSQ, Gif-Sur-Yvette 91191, France.

[3] Department of Biology and Graduate Degree Program in Ecology, Colorado State University, Fort Collins, CO 80523, USA.

[4] Department of Microbiology and Plant Biology, University of Oklahoma, Norman, OK 73019, USA.

[5] Institute of Ecology, University of Innsbruck, 6020 Innsbruck, Austria.

[6] Department of Biology, University of Antwerp, Universiteitsplein 1, 2610 Wilrijk, Belgium.

[7] Institute of Environmental Engineering, ETH Zurich, 8093 Zurich, Switzerland.

[8] Institute for Atmospheric and Climate Science, ETH Zurich, 8092 Zurich, Switzerland.

[9] National Institute for Environmental Studies, Tsukuba, Ibaraki 305-8506, Japan.

[10] Karlsruhe Institute of Technology, 82467 Garmisch-Partenkirchen, Germany.

[11] College of Engineering, Mathematics and Physical Sciences, University of Exeter, Exeter, EX4 4QF, UK.

[12] ARC Centre of Excellence for Climate System Science, University of New South Wales, Kensington, NSW 2052, Australia.

[13] Department of Civil and Environmental Engineering, Imperial College London, London, SW7 2AZ, UK.

[14] NASA Goddard Space Flight Center, Biospheric Sciences Laboratory, Greenbelt, MD 20771, USA.

[15] Institute of Environment Sciences, Biology Science Department, University of Quebec at Montreal, Montreal H3C 3P8, Quebec, Canada.

[16] State Key Laboratory of Soil Erosion and Dryland Farming on the Loess Plateau, College of Forestry, Northwest A & F University, Yangling 712100, China.

[17] Environmental Sciences Division and Climate Change Science Institute, Oak Ridge National Laboratory, Oak Ridge, Tennessee 37831-6301, USA.

[18] International Center for Climate and Global Change Research, School of Forestry and Wildlife Sciences, Auburn University, Auburn, AL 36849, USA.

[19] Institute of Arctic Biology, University of Alaska Fairbanks, Fairbanks, Alaska 99775, USA.

[20] Max Planck Institute for Meteorology, 20146 Hamburg, Germany.

[21] School of Geosciences, University of Edinburgh, Edinburgh EH9 3FF, UK.

[22] Research School of Biology, Australian National University, Canberra, ACT 2601, Australia.

[23] Norwegian Institute of Bioeconomy Research, 1431 Ås, Norway.

[24] Woods Hole Research Center, Falmouth, Massachusetts 02540-1644, USA.

[25] College of Life and Environmental Sciences, University of Exeter, Exeter EX4 4RJ, UK.

[26] CSIRO Oceans and Atmosphere, PMB #1, Aspendale, Victoria 3195, Australia.

[27] Center for Ecosystem Sciences and Society, Department of Biological Sciences, Northern Arizona University, Flagstaff, AZ 86011, USA.

*Correspondence to*: Donghai Wu (donghai.wu@pku.edu.cn)

**Abstract**

Field measurements of aboveground net primary productivity (ANPP) in temperate grasslands suggest that both positive and negative asymmetric responses to changes in precipitation may occur. Under normal range of precipitation variability, wet years typically result in ANPP gains being larger than ANPP declines in dry years (positive asymmetry), whereas increases in ANPP are lower in magnitude in extreme wet years compared to reductions during extreme drought (negative asymmetry). Whether the current generation of ecosystem models with a coupled carbon-water system in grasslands are capable of simulating these asymmetric ANPP responses is an unresolved question. In this study, we evaluated the simulated responses of temperate grassland primary productivity to scenarios of altered precipitation with fourteen ecosystem models at three sites, Shortgrass Steppe (SGS), Konza Prairie (KNZ) and Stubai Valley meadow (STU), spanning a rainfall gradient from dry to moist. We found that: (1) The spatial slopes derived from modeled primary productivity and precipitation across sites were steeper than the temporal slopes obtained from inter-annual variations, which was consistent with empirical data. (2) The asymmetry of the responses of modeled primary productivity under normal inter-annual precipitation variability differed among models, and the mean of the model-ensemble suggested a negative asymmetry across the three sites, which was contrary to empirical evidence based on filed observations. (3) The mean sensitivity of modeled productivity to rainfall suggested greater negative response with reduced precipitation than positive response to an increased precipitation under extreme conditions at the three sites. (4) Gross primary productivity (GPP), net primary productivity (NPP), aboveground NPP (ANPP) and belowground NPP (BNPP) all showed concave-down nonlinear responses to altered precipitation in all the models, but with different curvatures and mean values. Our results indicated that most models overestimate the negative drought effects and/or underestimate the positive effects 
[revised manuscript text omitted]

These three grasslands were selected because they lie along a mean annual precipitation (MAP) gradient, and have detailed meteorological data to force the models. While two are "natural" grasslands (KNZ and SGS) and one (STU) is not, global land surface models do not typically differentiate regarding the origin of ecosystem types and heavily managed grasslands and pastures represent a significant fraction of mesic grasslands globally. Semi-natural subalpine grasslands in the Alps were created several centuries ago, are very lightly managed and should be in equilibrium concerning soil physical conditions. It should be noted though that the grassland at STU is cut once a year and lightly fertilized every 2-4 years and in consequence

differs in plant composition and soil fungi: bacteria ratio, which leads to different drought responses compared to abandoned grassland (Ingrisch et al., 2017; Karlowsky et al., 2018). Further, it is worth noting that the mesic grassland in the USA would also be forested if human-initiated prescribed fires were to be removed from the system (Briggs et al. 2005). Thus, these grassland sites lie along a continuum of dry natural grassland, mesic natural grassland maintained by human management, and anthropogenic moist grassland maintained by human management.

**2.2 Ecosystem model simulations**

In order to test the hypothesis of an asymmetric response of productivity to variable rainfall (Knapp et al., 2017b), simulations were conducted with fourteen ecosystem models CABLE, CLM45-ORNL, DLEM, DOS-TEM, JSBACH, JULES, LPJ-GUESS, LPJmL-V3.5, ORCHIDEE-2, ORCHIDEE-11, T&C, TECO, TRIPLEX-GHG and VISIT all using the same protocol defined by the precipitation subgroup of the model-experiment interaction study (Table 2). At all three grassland sites, observed and altered multi-annual hourly rainfall forcing time series were combined with observations of other climate variables. These variables were air temperature, incoming solar radiation, air humidity, wind speed and surface pressure. Model simulations were carried out using soil texture properties measured at each site as reported in Table 1. Simulated productivity during the observational period is influenced at least in some models (for instance those having C-N interactions) by historical climate change and $CO_2$ changes since the pre-industrial period. Thus instead of assuming that productivity was in equilibrium with current climate, historical reconstructions of meteorological variables from gridded CRUNCEP data at 1/2 hourly time step (Wei et al., 2014) were combined and bias-corrected with site observations to provide bias corrected historical forcing time series from 1901 to 2013 (CRUNCEP-BC). In addition to the observed current climate defining the ambient simulation, nine altered rainfall forcing datasets were constructed by decreasing/increasing the amount of precipitation in each precipitation event by -80%, -70%, -60%, -50%, -20%, +20%, +50%, +100% and +200% during the time-span of productivity observations at each site, leaving all other meteorological variables unchanged and equal to the observed values. Modelers performed all simulations described below based on the same protocol (see below) and the model output was compared with measured ecosystem productivities (GPP, NPP, ANPP and BNPP), whenever available

-   Simulation S0 spin-up: models simulated an initial steady state spin-up run for water and biomass pools under pre-industrial conditions using the 1901-1910 CRUNCEP-BC climate forcing in a loop and applying fixed atmospheric $CO_2$ concentration at the 1850 level.

-   Simulation S1 historical simulation from 1850 until the first year of measurement (1986 for SGS, 1982 for KNZ, and 2009 for STU): starting from the spin-up state, models were prescribed with increasing atmospheric $CO_2$ concentrations and dynamic historical climate from CRUNCEP-BC. Because there is no CRUNCEP-BC data for 1850-1900, the CRUNCEP-BC climate data from 1901 to 1910 was repeated in a loop instead.

- Simulation SC1 ambient simulation for the measurement periods (1986-2009 for SGS, 1982-2012 for KNZ, and 2009-2013 for STU) with observed $CO_2$ concentrations and meteorological data corresponding to site observations at the hourly or half-hourly scale.

- Simulations SP1-SP9 altered precipitation simulations for the measurement periods (1986-2009 for SGS, 1982-2012 for KNZ, and 2009-2013 for STU), starting from the initial state in the start year of the period and run using the nine altered rainfall forcing datasets with observed $CO_2$ concentration.

**2.3 Metrics of the response of productivity to precipitation changes**

In the analysis, we begin with testing our first specific objective, i.e., if the productivity-P sensitivities of spatial relationships are greater than the temporal ones in the models as found in the observations. We calculated the temporal slopes and spatial slopes between productivities and precipitation from multi-year ambient simulations (SC1). Temporal slopes are site based and relate inter-annual variability in precipitation to inter-annual variability in the productivities using linear regression analysis. Spatial slopes relate mean annual precipitation to mean annual productivity across the three sites.

We then calculated two indices to analyze the asymmetric responses of primary productivity to precipitation simulated by ecosystem models and derived by observations whenever data were available. The two indices are: (1) the asymmetry of productivity-P for current inter-annual variability, based on SC1 where observations for ANPP are also available; and (2) the sensitivity of productivity to P for simulations where mean precipitation was altered, based on SP results. With these metrics, we test our second and third specific objectives, i.e., whether models could reproduce the observed asymmetric responses of productivity in grasslands to altered precipitation under normal and extreme conditions.

Finally, we analyze the nonlinearity of modeled response of productivity to precipitation, which is described by the parameters of the curvilinear productivity-P relationships across the full range of altered precipitation scenarios, based on fits to model output for the ambient (SC1) and altered (SP) simulations. Detailed methods for the two indices used to analyze the asymmetric responses of primary productivity to altered precipitation and the curvilinear productivity-P relationships are introduced in the following.

**2.3.1 Asymmetry index from inter-annual productivity and precipitation**

In order to characterize the asymmetry of productivity to precipitation, we define the asymmetry index (AI) from inter-annual productivity and precipitation data as follows:

$$AI = R_p - R_d \tag{1}$$

where $R_p$ is the relative productivity pulse in wet years, and $R_d$ is the relative productivity decline in dry years defined by:

$$R_p = (med(f_{p90}) - \bar{f})/\bar{f} \tag{2}$$

$$R_d = (\bar{f} - med(f_{p10}))/\bar{f} \tag{3}$$

where $f$ is the inter-annual productivity, being a function of environmental factors from models or observation; $\bar{f}$ is mean annual productivity in the period of measurements (Table 1); $med(f_{p90})$ is the median value of productivities in wet years with annual precipitation higher than the 90[th] percentile level; $med(f_{p10})$ is median value of productivities in all the dry years when annual precipitation is lower than the 10[th] percentile level.

In general, $R_p > 0$ indicates that the median value of productivities in wet years is higher than the mean annual productivity in the period of measurements; and $R_d > 0$ indicates that the median value of productivities in dry years is smaller than the mean annual productivity in the period of measurements. Therefore, if $AI > 0$, i.e., a positive asymmetry, means that there is a greater increase of productivity in wet years than decline in dry years; if $AI < 0$, i.e., a negative asymmetry, means that there is a greater decline of productivity in dry years than increase in wet years.

Furthermore, uncertainty ranges of $R_p$, $R_d$ and $AI$ were estimated as follows:

$$R_p \in \left[R_{p_{low}}, R_{p_{up}}\right] = \left[\frac{(med(f_{p90}) - mad(f_{p90})) - \bar{f}}{\bar{f}}, \frac{(med(f_{p90}) + mad(f_{p90})) - \bar{f}}{\bar{f}}\right] \tag{4}$$

$$R_d \in \left[R_{d_{low}}, R_{d_{up}}\right] = \left[\frac{\bar{f} - (med(f_{p10}) + mad(f_{p10}))}{\bar{f}}, \frac{\bar{f} - (med(f_{p10}) - mad(f_{p10}))}{\bar{f}}\right] \tag{5}$$

$$AI \in \left[AI_{low}, AI_{up}\right] = \left[R_{p_{low}} - R_{d_{up}}, R_{p_{up}} - R_{d_{low}}\right] \tag{6}$$

where $R_{p_{low}}$ and $R_{p_{up}}$ are the lower and upper bounds of $R_p$ using one median absolute deviation, i.e., $mad(f_{p90})$; $R_{d_{low}}$ and $R_{d_{up}}$ are the lower and upper bounds of $R_d$ using one median absolute deviation, i.e., $mad(f_{p10})$; $AI_{low}$ and $AI_{up}$ are the lower and upper bounds of $AI$ corresponding to estimated $R_p$ and $R_d$ ranges.

**2.3.2 Sensitivity of productivity to altered versus inter-annual precipitation variability**

For altered precipitation, in particular for the extreme SP simulations where mean precipitation was altered and annual precipitation of a few years was outside the range of observed precipitation variation, we tested the hypothesis whether the asymmetry response becomes negative, that is the impacts of extreme dry conditions on productivity are much greater than the positive effects of extreme wet scenarios (Knapp et al., 2017b). Thus, we tested the mean change in productivity imposed by the change in precipitation, and we defined the sensitivity of productivity to altered rainfall conditions (*S*) as:

$$S = (\overline{f_{P_a}} - \overline{f_{P_c}})/(|\bar{P}_a - \bar{P}_c|) \tag{7}$$

where $\overline{f_{P_a}}$ and $\overline{f_{P_c}}$ are the mean productivities of altered and ambient simulations; $\bar{P}_a$ and $\bar{P}_c$ are the mean annual precipitation amounts in altered and ambient simulations. It should be noted that the sensitivity of productivity to altered rainfall conditions could present the asymmetry response from normal to extreme conditions.

**2.3.3 Curvilinear productivity-P relationships across the entire range of altered P**

In general, plant productivity increases with increasing precipitation, and saturates when photosynthesis becomes less limited by water scarcity. We fitted the response of simulated productivity to altered precipitation using the Eq. (8):

$$y = a(1 - e^{-bx}) \tag{8}$$

5 Where the independent variable $x$ is the mean annual precipitation (mm), and the dependent variable $y$ one of the productivities (GPP, NPP, ANPP and BNPP). Parameter $a$ (g C m$^{-2}$ yr$^{-1}$) is the maximum value of productivity at high precipitation; and parameter $b$ (mm$^{-1}$) is the curvature of modeled productivity to altered precipitation.

**3 Results**

**3.1 Temporal versus spatial slopes of productivity-P**

10 From the ambient simulations, ensemble model results indicate that the slopes of the spatial relationships were steeper than the temporal slopes for GPP, NPP and ANPP for the subset of models that simulated this flux, while these differences in slopes were less obvious for BNPP (Fig. 1). We compared model results with site-observations for ANPP-P temporal slopes of the ambient simulation across the three sites (Fig. 1c). Observed and modeled temporal slopes decreased from dry (SGS) to moist (STU) site, from 0.10 g C m$^{-2}$ mm$^{-1}$ (0.05 to 0.14 for the 10$^{th}$ and 90$^{th}$ percentiles) to 0.05 g C m$^{-2}$ mm$^{-1}$ (-0.14 to 0.55 for the

15 10$^{th}$ and 90$^{th}$ percentiles) in the observations, and from 0.14 g C m$^{-2}$ mm$^{-1}$ (0.02 to 0.36 for the 10$^{th}$ and 90$^{th}$ percentiles) to 0.03 g C m$^{-2}$ mm$^{-1}$ (-0.04 to 0.29 for the 10$^{th}$ and 90$^{th}$ percentiles) for the model ensemble mean. Although there were some discrepancies in the range of spatial and temporal slopes across models (Fig. S1), the multi-model ensemble mean captured the key observation of steeper spatial than temporal slopes for ANPP (Fig. 1).

**3.2 Asymmetry of the inter-annual primary productivity response to precipitation**

20 The asymmetry of each model was diagnosed using the asymmetry index (AI; Eq. (1)), which showed large variation across models (Fig. 2, S2). Considering all the models as independent ensemble members, the mean AI of GPP and NPP showed significantly negative values at p < 0.1 level for SGS (ensemble value of $-0.11_{-0.31}^{0.12}$ and $-0.20_{-0.48}^{0.11}$ respectively with 10$^{th}$ and 90$^{th}$ percentiles). Hence, for SGS simulated declines of GPP and NPP in dry years were larger than the increases in wet years. For STU, the mean AI values were only slightly negative (ensemble value for GPP $-0.03_{-0.07}^{0.02}$ and for NPP $-0.04_{-0.09}^{0.01}$

25 with 10$^{th}$ and 90$^{th}$ percentiles), while AI was very close to zero at KNZ. By contrast, observation-based AI values, estimated from long-term inter-annual ANPP measurements, suggest a decrease from positive ($0.32_{0.14}^{0.49}$ for SGS and $0.20_{0.04}^{0.37}$ for KNZ) to negative (-0.21 for STU). At the dry (SGS) and mesic (KNZ) sites (Fig. S2), most of model simulations overestimated the extent of negative drought effects in dry years ($R_d$) and/or underestimated the positive impacts on ANPP in wet years ($R_p$). For example, CABLE and ORCHIDEE-2 overestimated the drought effects in dry years at both the two sites, and CLM45-ORNL

and VISIT underestimated the positive impacts in wet years at both the two sites (Fig. S2). At the moist site (STU), models agreed with observations regarding the negative sign of AI (negative asymmetry) but AI magnitude is not well captured.

**3.3 Sensitivities of primary productivity to altered precipitation**

The model-derived sensitivities given by Eq. (7) generally presented greater negative impacts of reduced precipitation than positive effects of increased precipitation under both normal (inter-annual) and extreme conditions (Fig. 3). The results also indicated that models represented a constant asymmetry pattern (negative asymmetry under normal and extreme conditions) across the full range of altered precipitation rather than a double asymmetry pattern (positive asymmetry under normal condition and negative asymmetry under extreme condition) established by Knapp et al. (2017b), which confirmed that models didn't capture the positive asymmetric responses of productivities to altered precipitation under normal conditions for the dry (SGS) and mesic (KNZ) sites.

Primary productivity at the dry site (SGS) was more sensitive to precipitation changes compared to the moist site (STU). Along with increases in precipitation, the largest sensitivity values were found for SGS (ensemble mean of $1.35_{0.42}^{2.49}$ g C m$^{-2}$ mm$^{-1}$ for GPP with 10$^{th}$ and 90$^{th}$ percentiles, $0.68_{0.24}^{1.47}$ g C m$^{-2}$ mm$^{-1}$ for NPP, $0.24_{0.08}^{0.61}$ g C m$^{-2}$ mm$^{-1}$ for ANPP and $0.16_{0.14}^{0.18}$ g C m$^{-2}$ mm$^{-1}$ for BNPP) and then KNZ ($0.32_{-0.09}^{1.23}$ g C m$^{-2}$ mm$^{-1}$ for GPP, $0.20_{-0.05}^{0.72}$ g C m$^{-2}$ mm$^{-1}$ for NPP, $0.13_{0.01}^{0.21}$ g C m$^{-2}$ mm$^{-1}$ ANPP and $0.06_{0.01}^{0.28}$ g C m$^{-2}$ mm$^{-1}$ for BNPP with 10$^{th}$ and 90$^{th}$ percentiles) when precipitation was altered by +20%. The values of S decreased with further increased precipitation, indicating that additional water does not increase productivity in the same proportion exceeding a certain threshold. In contrast to SGS, the values of sensitivity for both GPP and NPP at STU are close to zero in response to added precipitation conditions, implying that the precipitation above ambient was not a limiting factor for grassland production in the models at this site.

The values of sensitivity decreased with reduced precipitation at KNZ and SGS, indicating larger negative impacts on primary productivity when conditions become drier. For the moist site of STU, primary productivities showed less sensitivity to moderately dry conditions, and sensitivity only increased with more extreme rainfall alterations out of $3\sigma$ (~40% precipitation change). Additionally, the values of S for ANPP were smaller than those of BNPP at KNZ and SGS, while there were no differences between ANPP and BNPP at STU (Fig. 3). Thus, model results suggest that the dry site (SGS) can be particularly vulnerable to altered rainfall than the moist site (STU) which was more robust in response to altered rainfall.

**3.4 Curvilinear responses of productivity to altered precipitation**

At SGS and KNZ, simulated GPP and NPP increased with increasing precipitation. In contrast, at the moist STU, most models showed saturation in productivity for precipitation above ambient values (Fig. 4). Along with increasing precipitation, GPP and NPP showed nonlinear concave-down response curves in all models, with different curvatures *b* and maximum

[revised manuscript text omitted]

The sensitivity of productivity to increased and decreased precipitation for simulations where mean precipitation was normally altered generally suggested negative asymmetric responses at dry (SGS) and mesic (KNZ) sites (Fig. 3c). This contrasts with a meta-analysis of grassland precipitation manipulation experiments (Wilcox et al., 2017) and with the ANPP-P conceptual model (Knapp et al., 2017b), which suggest a positive asymmetry response in the range of normal rainfall variation. This emphasizes the finding that most models overestimate drought effects and/or underestimate wet year impacts on primary productivity of dry and mesic sites for current precipitation variability. Under extreme conditions with modified precipitation, models were in line with the hypothesis and the data showing that ANPP saturates in very wet conditions but declines strongly in very dry conditions (Knapp et al., 2017b). For BNPP sensitivities to altered precipitation, meta-analysis of previous experiments indicated symmetric responses to increasing and decreasing rainfall (Luo et al., 2017; Wilcox et al., 2017), which may be regulated by allocation controls on the ratio of ANPP and BNPP to total NPP in response to altered precipitation. However, in the participating models, BNPP shows a negative asymmetric responses to altered rainfall (Fig. 3d), which may reflect a shortcoming of carbon-water interactions in the belowground ecosystems.

**4.2 Curvilinear responses of productivities to altered precipitation by models**

In general, precipitation in ecosystem models is distributed through three pathways (Smith et al., 2014b): (1) intercepted by vegetation and subsequently evaporated or falling on the ground; (2) infiltrated into the upper soil layers with subsequent evaporation, root water uptake and plant transpiration, or percolated down to deeper layers to form ground water; (3) runoff from the soil surface if the intensity of precipitation exceeds infiltration rates. In reality as well as in models, soil moisture rather than precipitation is the variable regulating vegetation growth, and biological responses to changes in precipitation are manifested as functions of soil moisture in different soil layers (Sitch et al., 2003; Smith et al., 2014b; Vicca et al., 2012). We calculated the surface soil water content (SSWC, 0-20cm depth converted from reported soil layers) and total soil water content (TSWC) under ambient and altered precipitation as simulated by the fourteen models, and we found different patterns with parabolic, asymptotic and threshold-like nonlinear curves, which is similar to the response curves of primary productivity at the three sites (Fig. S5, S6). For the moist STU, SSWC and TWSC did not show obvious changes in response to increased precipitation since soil moisture at this site is often relatively near field capacity, while the SSWC and TSWC quickly decreased with decreasing in precipitation (Fig. S5, S6). In contrast, SSWC and TSWC at SGS showed significant increases in response to altered increased precipitation, and slow decreases for decreased precipitation, because the soil was already very dry under average ambient conditions. Thus, changes of SWC in response to precipitation contribute to driving the different response patterns of simulated primary productivity across the grassland sites.

The responses of primary productivity to precipitation in models might also be driven by the intrinsic structure and parameterizations of vegetation functioning besides changes of soil moisture (Gerten et al., 2008), which account for the large spread in the values of *b* and *a* among models at the three sites (Figure 4, 5, S3, S4). For example, carbon-nitrogen cycle coupling in ecosystem models reduced the simulated vegetation productivity relative to a carbon-only counterpart model (Thornton et al., 2007; Zaehle et al., 2010). Of those models used in this study, only five of the 14 models include carbon-nitrogen-water interactions (Table 2, S1, S2). We calculated the ensemble mean of productivity for this group of carbon-nitrogen models (CLM45-ORNL, DLEM, DOS-TEM, LPJ-GUESS and TRIPLEX-GHG) and carbon-only models (CABLE, JSBACH, JULES, LPJmL-V3.5, ORCHIDEE-2, ORCHIDEE-11, T&C, TECO and VISIT) across altered and ambient precipitation simulations at the three sites, and then fitted the productivity-P responses with Eq. (8) (Fig. S7, S8, S9). We found that ensemble mean of carbon-nitrogen models generally produce a weaker GPP, NPP and ANPP response to precipitation than ensemble mean of carbon-only models, and similar responses for BNPP. The latter may be explained by fixed root profiles in most models (Table S13). Our findings suggest that N interactions in ecosystem models reduced the productivity-P sensitivities, but should be confirmed using the same model prescribed with different N availability. In addition to the influence of nutrient cycling, different definitions of vegetation compositions (C3/C4) (Table S14), root profiles (Table S13), phenology (Table S9) and carbon allocation (Table S4) at the three sites may also contribute to the large variations of modeled productivity-P

responses and demands for more accurate calibration of models to the specificity of the local sites in future model intercomparison studies.

**4.3 Uncertainties, knowledge gaps and suggestions of further work**

In this work, we applied two indices to characterize the asymmetry responses in the normal precipitation range using inter-annual variability of present conditions and forcing models with continuously modified precipitation amounts. Asymmetry indices from the inter-annual gross and net primary productivities suggest large uncertainties (Fig. 2), while the sensitivity analysis to changes in mean precipitation reported clear responses (Fig. 3). This can be explained by the differences in other climatic factors (for example, temperature, radiation, and vapor pressure), or timing and frequency of precipitation between dry and wet years. All these uncontrolled factors may contribute to the large uncertainties of asymmetric responses from inter-annual variations (Chou et al., 2008; Peng et al., 2013; Robertson et al., 2009).

Although the carbon-water interactions in current models have been improved during the last decades, there still exist large gaps for accurately diagnosing the errors in the representation of key processes and parameterizations. Suggestions that should be considered in future studies aimed at model-data interaction include: (1) models should report SWC at the same depth of experiments and experimental data should be made available for better comparisons in following studies. This can provide insights into the bias of modeled sensitivities to precipitation and check explicitly the sensitivity of vegetation productivity to change in SWC; (2) more experiments are needed that assess also BNPP in order to evaluate the corresponding processes in models (Luo et al., 2017; Wilcox et al., 2017); (3) there still exist large gaps between changes of precipitation occurrence and intensity in reality and how we simulated them in the current work, i.e., the altered rainfall forcing datasets were constructed by decreasing/increasing the amount of precipitation in each precipitation event by a fixed percentage during the time-span of productivity observations at each site and not by modifying precipitation structure or reproducing the real treatment. Further studies need to consider better different scenarios of precipitation occurrence and intensity under climate change (Lauenroth and Bradford, 2012), which will likely help to better understand the responses of productivities to altered precipitation in the next decades. In addition, modelers will need to simulate the control experiments corresponding to the real local precipitation manipulations applied by field scientists, e.g., considering the observed time series of modified precipitation and vegetation composition, root profiles, nutrient cycling, phenology and carbon allocation as close as possible to local conditions. This should be a priority for future model-experiment interaction studies.

**5    Conclusions**

This is the first study where a large group of modelers simulated the response of grassland primary productivity to precipitation using long-term observations for evaluating the asymmetry responses to altered precipitation. Our results demonstrated that

multi-model ensemble mean captured the key observation of steeper spatial than temporal slopes for ANPP. On the other hand, our analyses revealed that most models do not capture the observed positive asymmetry responses for the dry (SGS) and mesic (KNZ) sites under the normal precipitation conditions, suggesting an overestimation of the drought effects and/or underestimation of the watering impacts on primary productivity in the normal state. In generally, current models represented a constant asymmetry pattern (negative asymmetry under normal and extreme conditions) across the full range of altered precipitation rather than a double asymmetry pattern (positive asymmetry under normal condition and negative asymmetry under extreme condition) established by Knapp et al. (2017b).

This study paves the path for further analyses where collaboration between modelers and site investigators needs to be strengthened such that also data other than ANPP can be considered and to identify which specific processes in ecosystem models are responsible for the observed discrepancies. This will eventually allow us to produce more reliable carbon-climate projections when facing different precipitation patterns in the future.

**Data availability**

All the modeled outputs in the first model-experiment interaction study can be publicly obtained from https://pan.baidu.com/s/1CXAnStQMBD_4a0tLGiIpiQ.

**Competing interests**

The authors declare that they have no conflict of interest.

**Acknowledgements**

This study was supported by National Natural Science Foundation of China (41530528). We also acknowledge support from the ClimMani COST action (ES1308). S.V. is a postdoctoral fellow of the Fund for Scientific Research - Flanders. M.K. acknowledges support from the EU FP7 project LUC4C, grant 603542. We thank Jeffrey S. Dukes, Shiqiang Wan and the organizers of the conference for model-experiment interaction study in Beijing. We thank Sibyll Schaphoff, Werner von Bloh, Susanne Rolinski and Kirsten Thonicke from PIK and Matthias Forkel from TU Vienna for their support of the LPJmL code. J. Mao, D. Ricciuto, and X. Shi were supported by the Terrestrial Ecosystem Science Scientific Focus Area (TES SFA) project funded through the Terrestrial Ecosystem Science Program in the Climate and Environmental Sciences Division (CESD) of the Biological and Environmental Research (BER) Program in the US Department of Energy Office of Science. The simulations of CLM4.5 used resources of the Oak Ridge Leadership Computing Facility at the Oak Ridge National Laboratory, which is supported by the Office of Science of the U.S. Department of Energy under Contract No. DE-AC05-00OR22725.

[revised manuscript text omitted]
 the model uncertainty range using interquartile spread of the temporal slopes between individual simulations (10[th] and 90[th] percentiles). The blue line is the ensemble mean of modeled productivities, and the blue error bar represents the model uncertainty range using interquartile spread of the productivities between individual simulations (10[th] and 90[th] percentiles). In (c), the grey lines are the observed temporal slopes, and the black line shows the observed spatial slope. The grey shading represents the observed uncertainty range using bootstrap sampling method (10[th] and 90[th] percentiles), and the black error bar represents the observed uncertainty range using interquartile spread of the inter-annual productivities (10[th] and 90[th] percentiles). Note that we simply converted observed ANPP from dry mass (g DM m$^{-2}$ yr$^{-1}$) to carbon mass (g C m$^{-2}$ yr$^{-1}$) with a factor of 0.5.

[Figure]

**Figure 2** Asymmetry responses of inter-annual GPP (a), NPP (b), ANPP (c) and BNPP (d) to precipitation in ambient simulations at the three sites SGS, KNZ and STU. The asymmetry index was calculated as the difference between the relative productivity pulses ($R_p$) and declines ($R_d$) in wet years and dry years (see Eq. (1) - Eq. (3)). Black pentagrams in (c) represent asymmetry indices from observations. The corresponding black error bars represent the observed uncertainty ranges using Eq. (4) - Eq. (6). A black asterisk at the bottom of a panel indicates a significant asymmetry response of the model ensemble at 0.1 significance level by a non-parametric statistical hypothesis test (Wilcoxon signed rank test).

[Figure]

**Figure 3** Sensitivity of GPP (a), NPP (b), ANPP (c) and BNPP (d) for altered precipitation simulations at the three sites SGS, KNZ and STU. Curves show the ensemble mean of models, and the shading represents the model uncertainty range using interquartile spread of the sensitivities between individual simulations (10th and 90th percentiles). Curves above the zero line represent responses under increasing precipitation conditions relative to the control, and curves below the zero line show responses under decreasing precipitation conditions relative to the control. Vertical dashed lines represent precipitation variations of one standard deviation (1$\sigma$), two standard deviations (2$\sigma$), and three standard deviations (3$\sigma$), which were derived from long-term annual precipitation at the three sites respectively.

[Figure]

**Figure 4** Responses of simulated annual GPP (left column), NPP (central column) and CUE (NPP / GPP; right column) to altered and ambient precipitation (P) levels at the three sites STU, KNZ and SGS. The fitted equation is Eq. (8) for GPP and NPP (see Fig. S3 for fitted *a* and *b*). The grey dashed line represents ambient precipitation. It should be noted that the x-axis scales are different between the sites.

[Figure]

**Figure 5** Responses of simulated annual ANPP (left column), BNPP (central column) and the ratio of ANPP and NPP (right column) to altered and ambient precipitation (P) levels at the three sites STU, KNZ and SGS. The fitted equation is Eq. (8) for ANPP and BNPP (see Fig. S4 for fitted *a* and *b*). The grey dashed line represents ambient precipitation. It should be noted that the x-axis scales are different between the sites.

**Table 1** Key plant, soil, and climate characteristics of the three grassland sites. MAT, mean annual temperature; and MAP, mean annual precipitation. MAT and MAP are based on the periods for the three sites with ANPP measurements.

| | SGS | KNZ | STU |
|---|---|---|---|
| Latitude | 40°49′ N | 39°05′ N | 47°07′ N |
| Longitude | 104°46′ W | 96°35′ W | 11°19′ E |
| MAT(℃) | 8.6±0.7 | 13.0±0.9 | 6.2±0.8 |
| MAP(mm yr$^{-1}$) | 304±118 | 827±175 | 1429±198 |
| ANPP (g DM m$^{-2}$ yr$^{-1}$) | 91±36 | 387±82 | 525±210 |
| Measurement period | 1986-2009 | 1982-2012 | 2009-2013 |
| Grassland type | Shortgrass steppe | Mesic tallgrass prairie | Subalpine meadow |
| C3 species (%) | 30 | 15 | 100 |
| C4 species (%) | 70 | 85 | 0 |
| Soil type | Aridic Argiustoll | Typic Argiustoll | Dystric Cambisol |
| Sand (%) | 14 | 8 | 42 |
| Silt (%) | 58 | 60 | 31 |
| Clay (%) | 27 | 32 | 27 |

**Table 2** Summary of ecosystem models used in this study, including model name, nitrogen (N) cycle and relevant references. Also see Table S1-S14 for details of the simulated processes for grasslands in the ecosystem models, including N cycle, phosphorus (P) cycle, carbon (C) allocation scheme, carbohydrate reserves, leaf photosynthesis and stomatal conductance including treatment of water stress, scaling of photosynthesis from leaf to canopy, phenology, mortality, soil hydrology, surface energy budget, root profile and dynamics, and grassland species.

| Model | Expanded Name | N cycle | References |
|---|---|---|---|
| CABLE | CSIRO Atmosphere Biosphere Land Exchange model | No | (Kowalczyk et al., 2006; Wang et al., 2011) |
| CLM45-ORNL | Version 4.5 of the Community Land Model | Yes | (Oleson et al., 2013) |
| DLEM | Dynamic Land Ecosystem Model | Yes | (Tian et al., 2011; Tian et al., 2015) |
| DOS-TEM | Dynamic Organic Soil structure in the Terrestrial Ecosystem Model | Yes | (Yi et al., 2010; McGuire et al., 1992) |
| JSBACH | Jena Scheme for Biosphere-Atmosphere Coupling in Hamburg | No | (Kaminski et al., 2013; Reick et al., 2013) |
| JULES | Joint UK Land Environment Simulator | No | (Best et al., 2011; Clark et al., 2011) |
| LPJ-GUESS | Lund-Potsdam-Jena General Ecosystem Simulator | Yes | (Smith et al., 2001; Smith et al., 2014a) |
| LPJmL-V3.5 | Lund-Potsdam-Jena managed Land | No | (Bondeau et al., 2007) |
| ORCHIDEE-2 | Organizing Carbon and Hydrology in Dynamic Ecosystems (2 soil layers) | No | (Krinner et al., 2005) |
| ORCHIDEE-11 | Organizing Carbon and Hydrology in Dynamic Ecosystems (11 soil layers) | No | (Krinner et al., 2005) |
| T&C | Tethys-Chloris | No | (Fatichi et al., 2012; Fatichi et al., 2016) |
| TECO | process-based Terrestrial Ecosystem model | No | (Weng and Luo, 2008) |
| TRIPLEX-GHG | An integrated process model of forest growth, carbon and greenhouse gases | Yes | (Peng et al., 2002; Zhu et al., 2014) |
| VISIT | Vegetation Integrative Simulator for Trace gases model | No | (Inatomi et al., 2010; Ito, 2010) |